# Generation of T cells with reduced off-target cross-reactivities by engineering co-signalling receptors

Jose Cabezas-Caballero [1], Anna Huhn[1], Mikhail A. Kutuzov [1], Violaine Andre[1], Alina Shomuradova[1], Bas W. A. Peeters [2], Geraldine M. Gillespie[2], P. Anton van der Merwe [1] & Omer Dushek [1] ✉

Adoptive T cell therapy using T cells engineered with novel T cell receptors (TCRs) targeting tumour-specific peptides is a promising immunotherapy. However, these TCR-T cells can cross-react with off-target peptides, leading to severe autoimmune toxicities. Current efforts focus on identifying TCRs with reduced cross-reactivity. Here we show that T cell cross-reactivity can be controlled by the co-signalling molecules CD5, CD8 and CD4, without modifying the TCR. We find the largest reduction in cytotoxic T cell cross-reactivity by knocking out CD8 and expressing CD4. Cytotoxic T cells engineered with a CD8→CD4 co-receptor switch show reduced cross-reactivity to random and positional scanning peptide libraries, as well as to self-peptides, while maintaining their on-target potency. Therefore, co-receptor switching generates super selective T cells that reduce the risk of lethal off-target cross-reactivity and offers a universal method to enhance the safety of T cell immunotherapies for potentially any TCR.

A promising immunotherapy approach is the adoptive transfer of T cells engineered with novel T cell receptors (TCR-T) recognizing tumour peptide antigens displayed on major histocompatibility complexes (pMHCs)[1]. This therapeutic strategy enables targeting nearly all tumour antigens, including tumour-specific developmental antigens and neo-antigens[2]. However, the engineered T cells can cross-react with off-target peptides in healthy tissues and cause fatal autoimmune toxicities[3–5]. This cross-reactivity has hampered efforts to produce highly potent TCR-T cell therapies[6,7].

Identifying the potential off-target cross-reactivities of TCRs before first-in-human clinical trials is challenging due to the lack of animal models or cell lines that represent the entire human proteome and human leucocyte antigen (HLA) allele diversity. Indeed, the clinical a3a TCR targeting the cancer-testis antigen MAGE-A3 passed safety screens but ultimately cross-reacted with a lower-affinity off-target peptide from the cardiac protein Titin, causing the death of two patients[4,5]. As a result, efforts are underway to establish pipelines to

identify effective yet safe TCRs[8–15]. Typically, these methods screen TCRs with different complementarity-determining regions (CDRs) for their ability to recognize the on-target tumour but not off-target self pMHCs[16]. In addition to screening methods, it has also been proposed that modifying the CDR loops to reduce their flexibility or introduce catch bonds may generally increase TCR specificity[17–19]. However, these strategies that rely on mutating the TCR sequence to reduce cross-reactivity require previous knowledge of the self-antigen that causes lethal cross-reactivity, and modifying the TCR sequence to reduce cross-reactivity to one antigen may result in new cross-reactivities to other self-antigens. Collectively, this makes it challenging and costly to screen and optimize each new candidate therapeutic TCR.

Instead of modifying the TCR CDR loops to reduce binding cross-reactivity, we hypothesized that functional cross-reactivity can be reduced by manipulating T cell signalling without modifying the TCR. In this way, even though the TCR can bind a large number

[1]Sir William Dunn School of Pathology, University of Oxford, Oxford, UK. [2]Nuffield Department of Medicine, University of Oxford, Oxford, UK. ✉e-mail: omer.dushek@path.ox.ac.uk

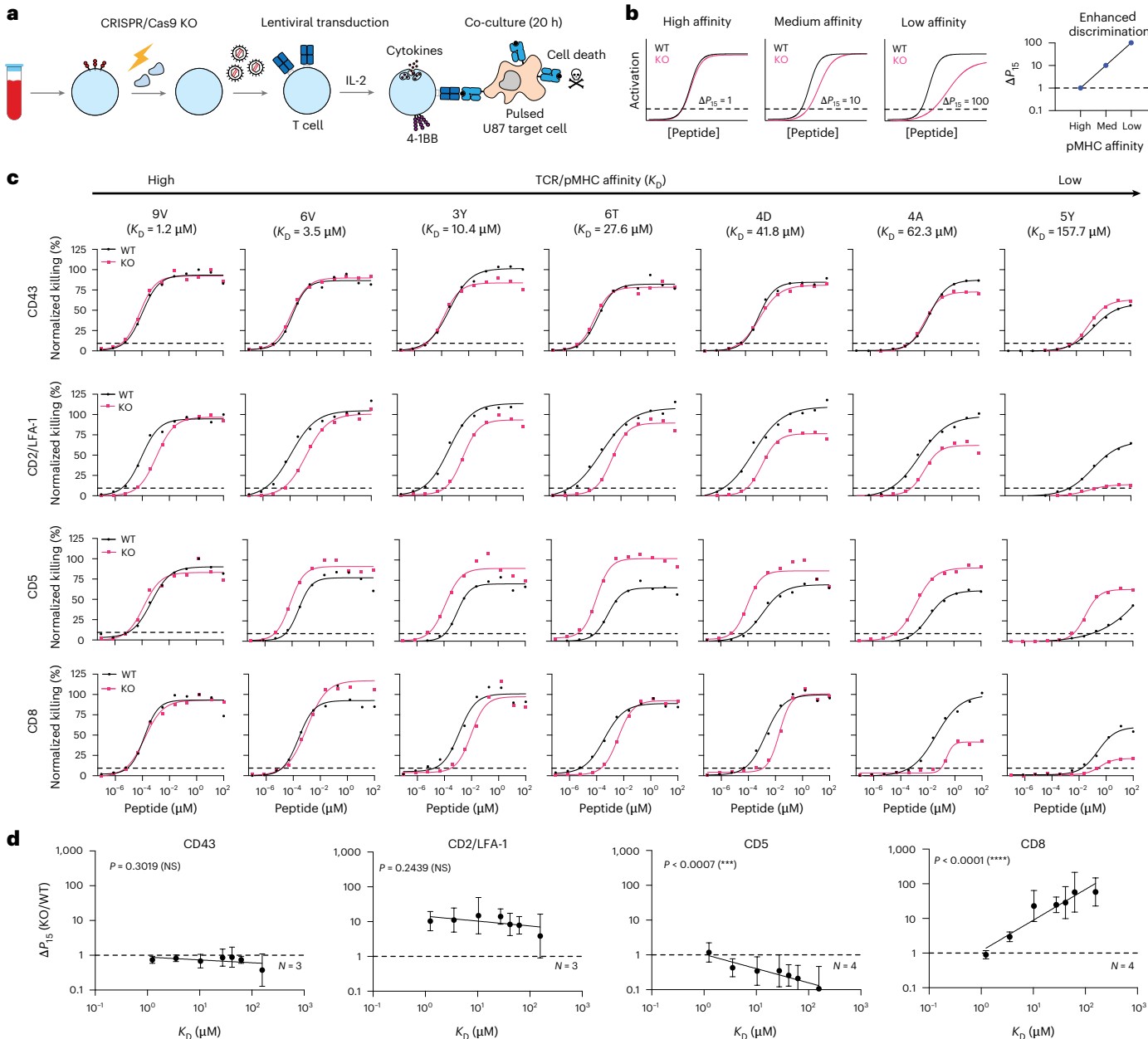

**Fig. 1 | Measuring the impact of different T cell co-signalling receptors on ligand sensitivity and discrimination. a**, Experimental workflow to produce gene knockout primary human c259 TCR-T cells. **b**, Schematic of analysis method to determine the impact of gene knockout on ligand discrimination: changes in ligand potency between WT and KO TCR-T cells are plotted for different ligand affinities. Ligand potency ($P_{15}$) is the ligand concentration required to activate 15% of maximum response. **c**, U87 cells were titrated with each of the 7 NY-ESO-1 peptides to stimulate WT or KO c259 TCR-T cells. Killing of the target U87 cells was measured after 20 h. Dashed line indicates potency ($P_{15}$). Data in **c** are representative of at least $N$ = 3 independent experiments with different blood donors. **d**, Fold change in $P_{15}$ between KO and WT T cells from **c** plotted over the TCR–pMHC affinity ($K_D$). Dashed line indicates fold change of 1. Data in **d** are shown as mean ± s.d. Significance of non-zero slope was assessed using a two-tailed $F$-test. NS, not significant; ***$P$ < 0.001, ****$P$ < 0.0001.

of peptides, T cells would only become activated in response to the few peptides that bind with high affinity. Put differently, we suggest that enhancing the ability of T cells to discriminate antigens based on their affinity would reduce their functional cross-reactivity. Given that co-signalling receptors on the T cell surface are known to impact TCR signalling[20], we reasoned that they impact T cell cross-reactivity.

Here we established a platform to quantify the impact of co-signalling receptors on T cell ligand discrimination. While a knock-out (KO) of the surface molecule CD5 decreased antigen discrimination, we found that a knockout of CD8 or expression of CD4 increased it. The largest effect was observed by combining CD8 knockout and CD4 expression ('co-receptor switch'). We demonstrate that a CD8→CD4 co-receptor switch dramatically reduced T cell cross-reactivity to peptide libraries and self-peptides. Overall, co-receptor switching is a broadly applicable strategy to produce super selective T cells that minimize the risk of lethal cross-reactivities without compromising on-target potency and can be potentially applied to any TCR.

## Results

### T cell co-signalling receptors differentially modulate ligand sensitivity and discrimination

We established a platform to quantify the contribution of different T cell co-signalling receptors to ligand discrimination. We selected the NY-ESO-1 specific c259 TCR contained in the investigational TCR-T

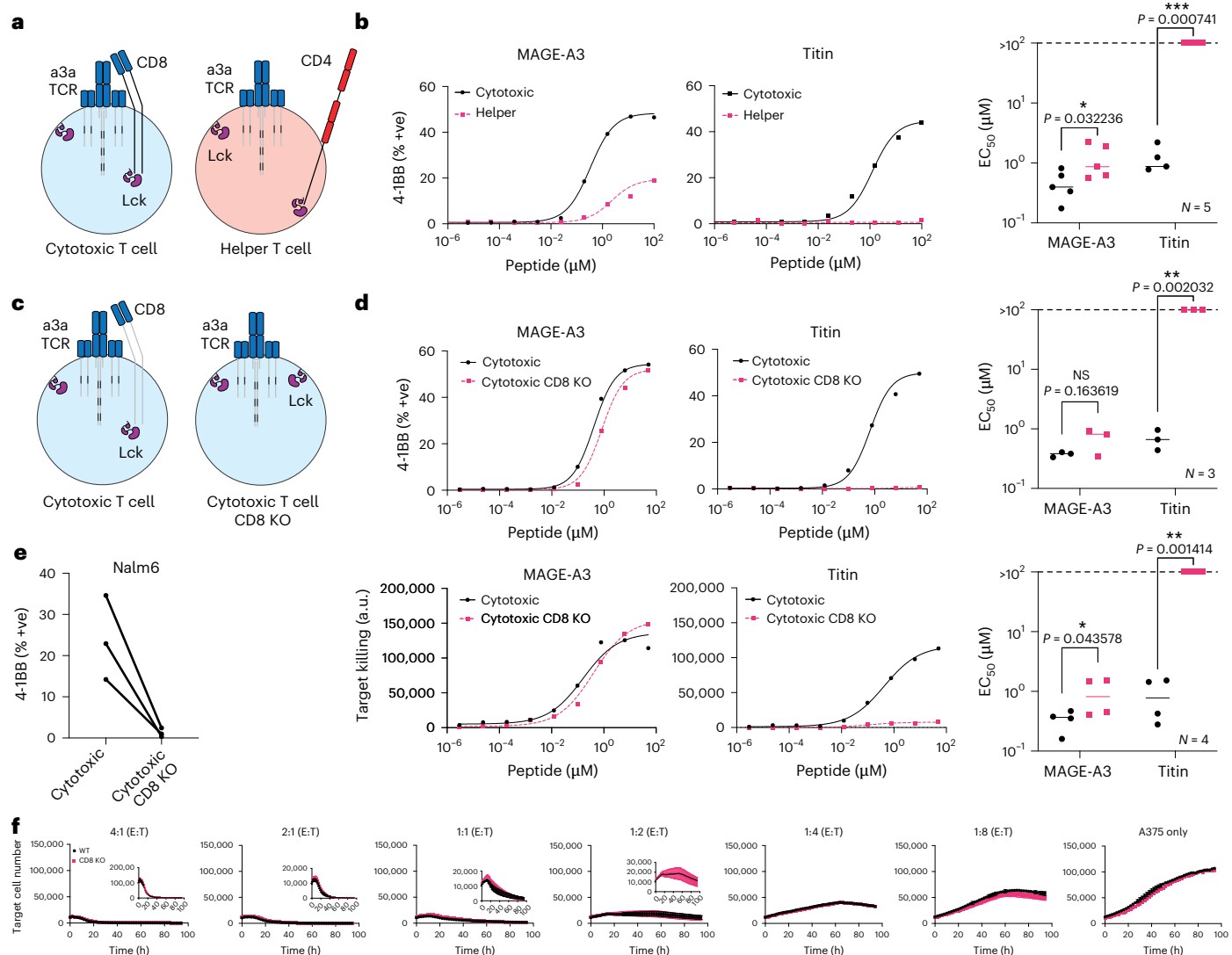

**Fig. 2 | CD8 co-receptor KO abolishes MAGE-A3 TCR cross-reactivity to the self-antigen Titin. a**, Schematic of helper and cytotoxic T cells transduced with the MAGE-A3-specific a3a TCR. **b**, HLA-A1+ T2 cells were titrated with MAGE-A3 or Titin peptides to stimulate cytotoxic or helper a3a TCR-T cells for 20 h. Representative dose–response curves (left, middle) and mean sensitivity as EC$_{50}$ (right). **c**, Schematic of WT or CD8 KO cytotoxic a3a TCR-T cells. **d**, HLA-A1+ T2 cells were titrated with MAGE-A3 or Titin peptides to stimulate WT or CD8 KO cytotoxic a3a TCR-T cells for 20 h. Representative dose–response curves (left, middle) and mean sensitivity as EC$_{50}$ (right). Data measuring 4-1BB surface activation marker (top) and target cell killing (bottom) are shown. **e**, Nalm6 cells endogenously expressing the Titin protein were co-cultured with WT or CD8 KO cytotoxic a3a TCR-T cells for 20 h. 4-1BB was stained by flow cytometry. **f**, A375

cells endogenously expressing the MAGE-A3 protein were co-cultured with WT or CD8 KO cytotoxic a3a TCR-T cells at different effector:target (E:T) ratios. A375 cell number was measured every 2 h. Inset plots show the same data over a reduced y-axis range to appreciate cell killing. Titrations in **b** and **d** are representative of at least $N = 3$ independent experiments with different blood donors. Each data point in **b** and **d** EC$_{50}$ plots represents an independent experiment with different blood donors. $P$ values were determined using two-tailed paired $t$-test; *$P < 0.05$, **$P < 0.01$, ***$P < 0.001$. Each data point in **e** represents an independent experiment with different blood donors. Data in **f** are shown as mean ± s.d. of technical triplicates. Representative data shown from $N = 3$ independent experiments with different blood donors.

therapy lete-cel as a model system[21]. The c259 TCR recognizes a 9-mer NY-ESO-1 peptide (SLLMWITQC) presented on HLA-A*02:01. It has previously been shown that a mutation of the anchor residue at the 9th position from cysteine to valine increases peptide stability on the MHC[22,23]. To enable the controlled study of TCR/pMHC binding, we therefore decided to use this heteroclitic peptide (9V, SLLMWITQ**V**). First, we measured the binding affinity of the c259 TCR to a panel of 7 variants of the 9V pMHC by surface plasmon resonance (SPR)[24] (Extended Data Fig. 1 and Supplementary Table 1). Second, we used CRISPR/Cas9 to knockout out five co-signalling receptors in primary human T cells expressing the c259 TCR that were previously suggested to impact ligand discrimination: CD8 (ref. 25), CD5 (ref. 26), CD43 (ref. 27), CD2 and LFA-1 (ref. 24,28) (Fig. 1a). Third, we co-cultured these

T cells with antigen-presenting cells (APCs) pulsed with a titration of each of the 7 peptides with different affinities to the TCR and assessed their ability to induce multiple measures of T cell activation (target cell killing, IFNγ secretion and 4-1BB upregulation) (Extended Data Figs. 2 and 3 and Supplementary Fig. 1). Finally, we quantified pMHC potency as the concentration of peptide required to elicit 15% activation ($P_{15}$) from WT or KO T cells (Fig. 1c, dashed horizontal line, see Methods for details). By plotting the fold change in potency ($\Delta P_{15}$) over affinity we could determine whether the co-signalling molecule was selectively decreasing activation to lower-affinity ligands (Fig. 1b).

We achieved high knockout efficiency of each co-signalling receptor (Extended Data Fig. 2a), enabling assessment of their impact on ligand discrimination (Fig. 1c,d, Extended Data Figs. 2 and 3

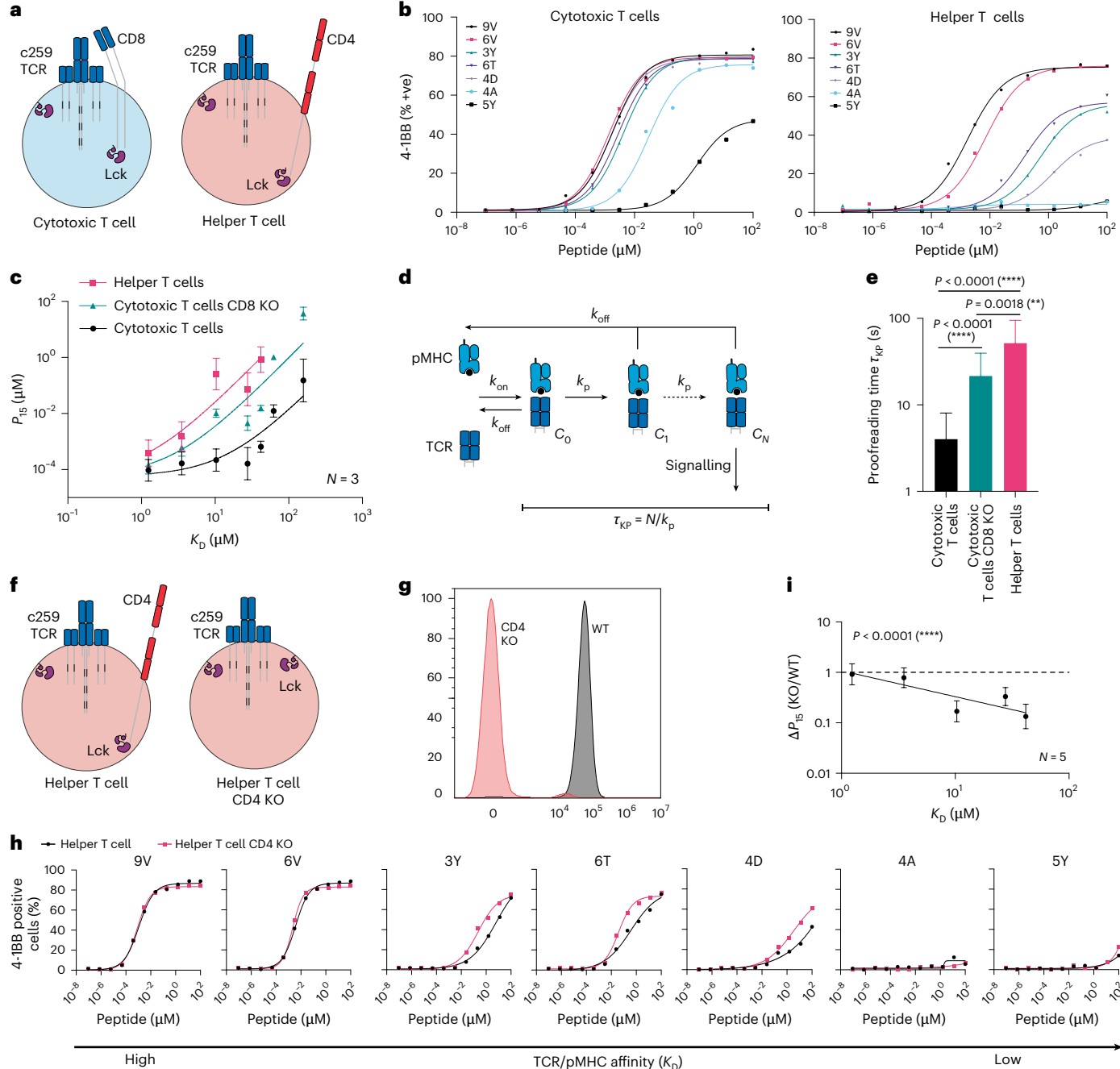

**Fig. 3 | The CD4 co-receptor enhances the discrimination of helper cells expressing an MHC-I-restricted TCR. a**, Helper and cytotoxic T cells express the CD4 and CD8 co-receptors, respectively. **b**, Representative ligand discrimination assays using helper and cytotoxic c259 TCR-T cells recognizing peptides on U87 target cells. Expression of the 4-1BB activation marker was measured after a 20 h co-culture. **c**, $P_{15}$ (mean ± s.d.) over TCR–pMHC affinity ($K_D$) from $N = 3$ independent blood donors (points) is fitted to the kinetic proofreading model (solid line). **d**, Kinetic proofreading introduces a time delay ($\tau_{KP}$) between pMHC binding (state $C_0$) and TCR signalling (state $C_N$) that selectively reduces signalling to low-affinity ligands. **e**, Fitted time delay from the data in **c** with 95% confidence interval. Two-tailed $F$-test compares the fitted proofreading time between conditions. **f**, Schematic of helper WT and CD4 KO c259 TCR-T cells. **g**, Flow cytometry staining of CD4 in WT and CD4 KO helper T cells. **h**, U87 cells were titrated with each of the 7 NY-ESO-1 peptides to stimulate WT or CD4 KO helper c259 TCR-T cells. 4-1BB expression was measured after 20 h. **i**, Fold change in potency ($P_{15}$) between CD4 KO and WT helper T cells from **h** is plotted over TCR–pMHC affinity ($K_D$). Dashed line indicates fold change of 1. Significance of non-zero slope was assessed using a two-tailed $F$-test. Data in **b**, **g** and **h** are representative of at least $N = 3$ independent experiments with different blood donors. Data in **i** are shown as mean ± s.d. of $N = 5$ independent experiments with different blood donors. **$P < 0.01$, ****$P < 0.0001$.

and Supplementary Fig. 1). The knockout of CD43 had no impact on activation, whereas the knockout of CD2 or LFA-1 individually or in combination reduced activation for all pMHC affinities to a similar extent and therefore, these molecules do not impact ligand discrimination. In contrast, a knockout of CD5 selectively improved activation

against lower-affinity ligands and therefore, CD5 KO reduced ligand discrimination. The knockout of CD8 selectively reduced activation to lower-affinity peptides without impacting the higher-affinity NY-ESO-1 9V peptide and therefore, CD8 KO increases ligand discrimination. Since the c259 TCR is affinity matured[29], we confirmed that

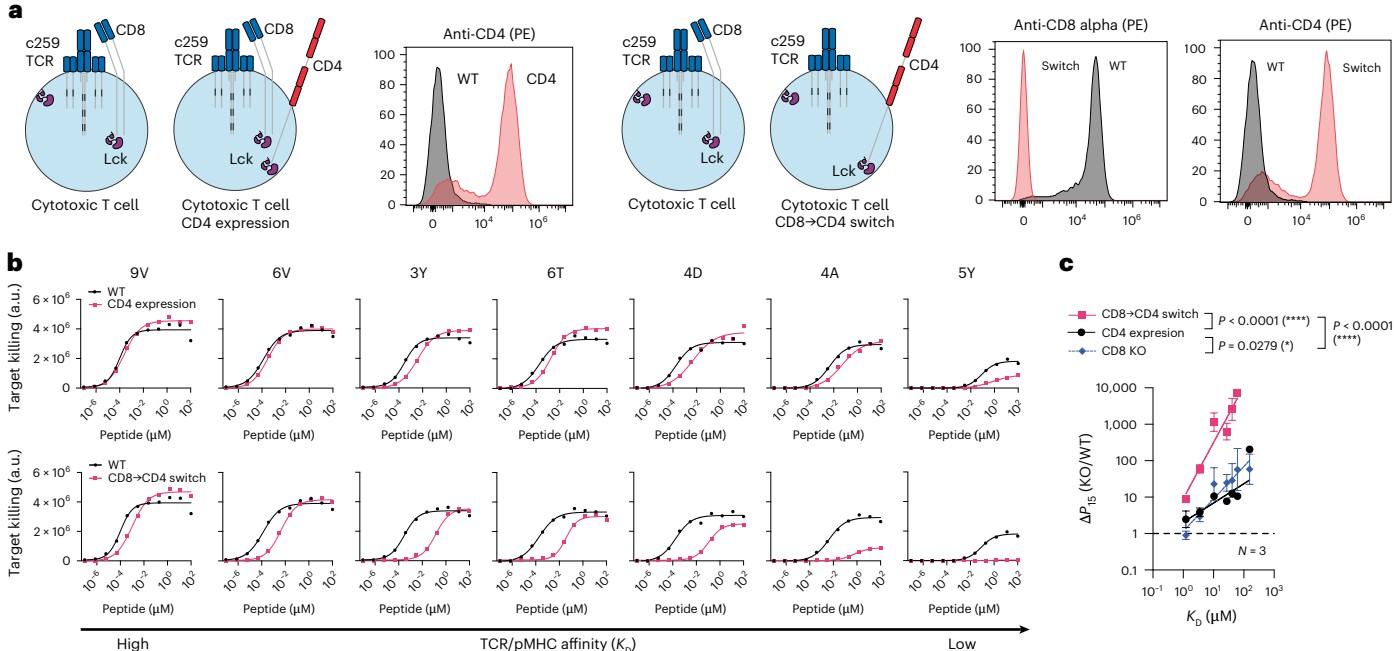

**Fig. 4 | Expression of the incompatible CD4 co-receptor in cytotoxic T cells enhances ligand discrimination. a**, Left: schematic of CD4 expression in cytotoxic T cells and flow cytometry staining of CD4 expression. Right: schematic of CD8→CD4 co-receptor switch T cells and flow cytometry staining of CD4 and CD8 expression. **b**, U87 cells were titrated with each of the 7 NY-ESO-1 peptides to stimulate WT or CD4-expressing cytotoxic T cells (top), or WT or CD8→CD4 co-receptor switch cytotoxic T cells (bottom). Target killing was measured after 20 h. **c**, The fold change in potency ($P_{15}$) between the indicated modified and WT cytotoxic T cells over TCR–pMHC affinity ($K_D$). Data for CD8 KO are shown from Fig. 1. Data in **a** and **b** are representative of $N = 3$ independent experiments with different blood donors. Data in **c** are shown as mean ± s.d. of $N = 3$ independent experiments with different blood donors. $P$ values were determined using a two-tailed $F$-test. *$P < 0.05$, ****$P < 0.0001$.

CD8 KO also increased the discrimination of the parental wild-type 1G4 TCR[23] (Supplementary Fig. 2). We observed lower maximum responses for lower-affinity ligands, and this observation is consistent with models of ligand discrimination[30,31]. Taken together, co-signalling molecules can control TCR ligand discrimination and a CD8 KO in particular can selectively reduce activation to lower-affinity ligands without impacting potency to the higher-affinity on-target antigen.

T cells engineered to express a new TCR still express their own endogenous TCR alpha and beta chains, which could mispair with the transgenic TCR chains and potentially affect T cell antigen recognition and consequently, affect ligand discrimination. We confirmed that deletion of the endogenous TCR α and β chains, using previously tested single guide (sg)RNAs targeting the TRAC/TRBC loci[32,33], did not impact ligand discrimination (Extended Data Fig. 4).

### CD8 knockout abolishes therapeutic a3a TCR cross-reactivity to Titin

T cells engineered with the MAGE-A3-specific a3a TCR caused lethal cardiac toxicities in a clinical trial due to cross-reactivity to a lower-affinity peptide from the muscle protein Titin[4,5]. Since we have demonstrated that the CD8 co-receptor can decrease T cell ligand discrimination, we decided to investigate whether the cross-reactivity to Titin was CD8 dependent.

Given that TCR-T therapies rely on expressing the therapeutic MHC-I-restricted TCR in both CD8+ cytotoxic and CD4+ helper T cells, we first examined their individual abilities to react to each antigen. While both cytotoxic and helper T cells responded to the on-target MAGE-A3 antigen, only cytotoxic T cells responded to the off-target Titin antigen, confirming that cytotoxic T cells are the likely source of autoimmune toxicity (Fig. 2a,b). By knocking out CD8 in cytotoxic T cells, we abolish activation against Titin without impacting responses to the higher-affinity on-target antigen (Fig. 2c,d). The CD8 KO also abolished the activation of T cells against Nalm6 cells that endogenously express

Titin[5] (Fig. 2e). Furthermore, in a longitudinal killing assay, we show that the knockout of CD8 in cytotoxic a3a TCR-T cells did not reduce their ability to kill the A375 melanoma cell line, which endogenously expresses MAGE-A3 (Fig. 2f and Supplementary Fig. 3).

### Helper T cells display enhanced discrimination against pMHC-I antigens due to their incompatible CD4 co-receptor

The observation that CD4+ helper T cells only responded to the higher-affinity MAGE-A3 antigen whereas CD8+ cytotoxic T cells also responded to the lower-affinity Titin antigen (Fig. 2a,b) suggested that helper T cells may have a different capacity to discriminate ligands.

We compared ligand discrimination in helper vs cytotoxic T cells using the NY-ESO-1 c259 TCR platform (Fig. 3a). Consistent with the a3a TCR, we found that cytotoxic T cells activated more strongly against low-affinity pMHCs than helper T cells (Fig. 3b,c). Interestingly, helper T cells displayed even higher discrimination than CD8 KO cytotoxic T cells (Fig. 3c).

The degree to which T cells are able to respond to lower-affinity antigens is partly determined by a kinetic proofreading mechanism that introduces a time delay between pMHC binding and TCR signalling[24,34] (Fig. 3d). This time delay is thought to be determined by biochemical steps that follow pMHC binding, including phosphorylation of ITAMs and ZAP70 by Lck, ZAP70 auto-phosphorylation, and the bridging of ZAP70 and LAT by Lck[35–37]. By fitting the proofreading model directly to the potency over pMHC affinity data (Fig. 3e), we confirmed that the time delay for helper T cells is even larger than for CD8 KO cytotoxic T cells. Thus, high levels of ligand discrimination for helper T cells cannot be explained simply by the absence of CD8 co-receptor alone.

Helper T cells express the CD4 co-receptor that, like CD8, has an intracellular association with Lck, but unlike CD8, cannot bind the MHC-I antigens targeted by the c259 TCR. We hypothesized that the presence of the incompatible CD4 co-receptor could be responsible for the enhanced discrimination of helper T cells by sequestering Lck from

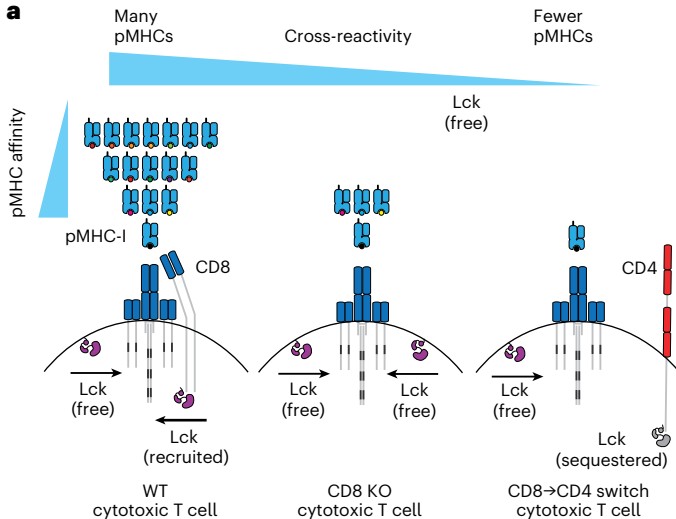

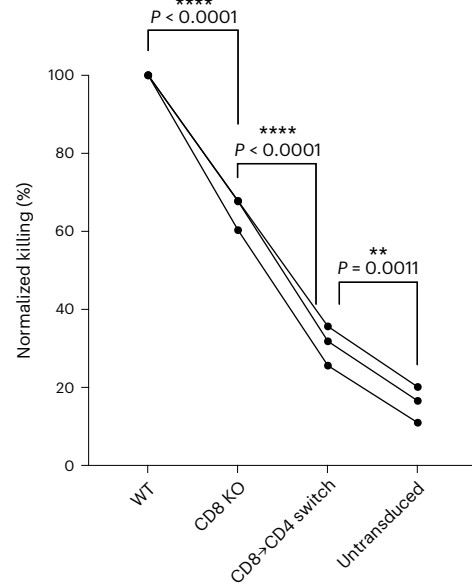

**Fig. 5 | CD8→CD4 co-receptor switch cytotoxic T cells display reduced cross-reactivity to a random 9-mer peptide library. a**, Schematic of the predicted cross-reactivity of WT, CD8 KO or CD8→CD4 co-receptor switch cytotoxic T cells. **b**, U87 cells were loaded with a randomly synthetized pool of 9-mer peptides (containing all natural amino acids except cysteine, with a theoretical diversity of $19^9$) to stimulate WT, CD8 KO or CD8→CD4 co-receptor switch cytotoxic c259 TCR-T cells. Target killing was measured after 20 h. Each data point represents an independent experiment with different blood donors. *P* values were determined using one-way analysis of variance (ANOVA) and Tukey's multiple-comparisons test. ***P* < 0.01, *****P* < 0.0001.

the TCR. Indeed, CD4 KO helper T cells displayed improved activation to lower-affinity peptides, reducing ligand discrimination compared with wild-type helper T cells (Fig. 3f–i). Therefore, the incompatible CD4 co-receptor increases the ligand discrimination of helper T cells targeting pMHC-I antigens. It follows that direct inhibition of Lck would also increase ligand discrimination, and we confirmed this by treating c259 TCR-T cells with a selective Lck inhibitor (Extended Data Fig. 5).

### Expression of the incompatible CD4 co-receptor in cytotoxic T cells enhances their ligand discrimination

Since the CD4 co-receptor increased the ability of helper T cells to discriminate ligands using an MHC-I-restricted TCR, we examined

whether it could also do this in cytotoxic T cells (Fig. 4a). Indeed, expression of CD4 in cytotoxic T cells selectively reduced activation and target killing against lower-affinity pMHCs, without affecting responses to the high-affinity 9V peptide (Fig. 4b and Extended Data Fig. 6a). Moreover, expression of CD4 in CD8 KO cytotoxic T cells synergized to produce T cells with extremely high levels of discrimination (Fig. 4b,c and Extended Data Fig. 6b,c). For example, whereas wild-type T cells can respond to the lower-affinity 4D peptide, these CD8→CD4 co-receptor switch T cells ignore this same antigen unless its concentration was increased by a dramatic ~3,000-fold. Thus, a CD8→CD4 co-receptor switch dramatically increased the ligand discrimination of cytotoxic T cells.

We also tested the impact of a CD8→CD4 co-receptor switch on activation against the wild-type NY-ESO-1 peptide (9C: SLLMWITQC). We find that the c259 TCR recognizes this ligand with a 5.5-fold weaker affinity than the 9V peptide variant (Extended Data Fig. 7a). Consequently, and in contrast to the 9V peptide, we find that the CD8→CD4 co-receptor switch shows a reduction in potency against U87 cells pulsed with the 9C peptide (Extended Data Fig. 7b,c) and also against A375 cells endogenously expressing NY-ESO-1 9C (Extended Data Fig. 7d).

We next examined the impact of a CD8→CD4 co-receptor switch on the autoimmune 1E6 TCR, which binds a peptide from preproinsulin with weak affinity and a peptide from *Clostridium asparagiforme* with intermediate affinity[38]. We found that a CD8→CD4 co-receptor switch increased ligand discrimination and maintained responses to the higher-affinity peptide (Extended Data Fig. 8).

### CD8→CD4 co-receptor switch cytotoxic T cells display reduced cross-reactivity while maintaining potent target killing

We next used three methods to examine how the increase in ligand discrimination that we report impacts T cell cross-reactivity.

In a pooled peptide library that contains a random mixture of peptides, it is expected that the majority of peptides that bind one TCR would do so with low affinity. As a result, we predicted that increasing ligand discrimination would reduce T cell cross-reactivity to a random pooled peptide library (Fig. 5a). We stimulated T cells with target cells pulsed with a random pooled 9-mer peptide library, where each position can be any amino acid except cysteine, with a theoretical diversity of $19^9$ peptides. Cytotoxic T cells expressing the c259 TCR killed target cells pulsed with the random peptide mixture, but reduced cross-reactive killing was observed in CD8 KO and especially in CD8→CD4 co-receptor switch T cells (Fig. 5b).

A positional scanning library includes all single amino acid changes relative to a target peptide (163 NY-ESO-1 variant peptides in the present case). Although cytotoxic c259 TCR-T cells killed targets expressing many of these peptides, CD8 KO cells and CD8→CD4 co-receptor switch T cells display reduced killing to many of these peptides except for the target peptide (Fig. 6a). To confirm that this reduced cross-reactivity was a result of increased ligand discrimination based on affinity, we developed a workflow to use a high-throughput SPR-based instrument to accurately and rapidly measure all 163 TCR/pMHC affinities (Fig. 6b, Extended Data Fig. 9 and Supplementary Table 2). As predicted, the reduced cross-reactivity of CD8 KO and CD8→CD4 co-receptor switch T cells to different concentrations of this peptide library was dependent on affinity, with reduced responses observed only to lower-affinity interactions (Fig. 6c,d and Supplementary Fig. 4).

Data from positional scanning libraries can also be used to predict TCR off-target cross-reactivities and this method was previously used to predict self-peptides from the human proteome recognized by the c259 TCR[8]. We screened these predicted self-peptides for their ability to activate c259 TCR-T cells, and for those that activated, we measured their affinity by SPR (Extended Data Fig. 10a–c and Supplementary Table 3). The CD8 KO and especially the CD8→CD4

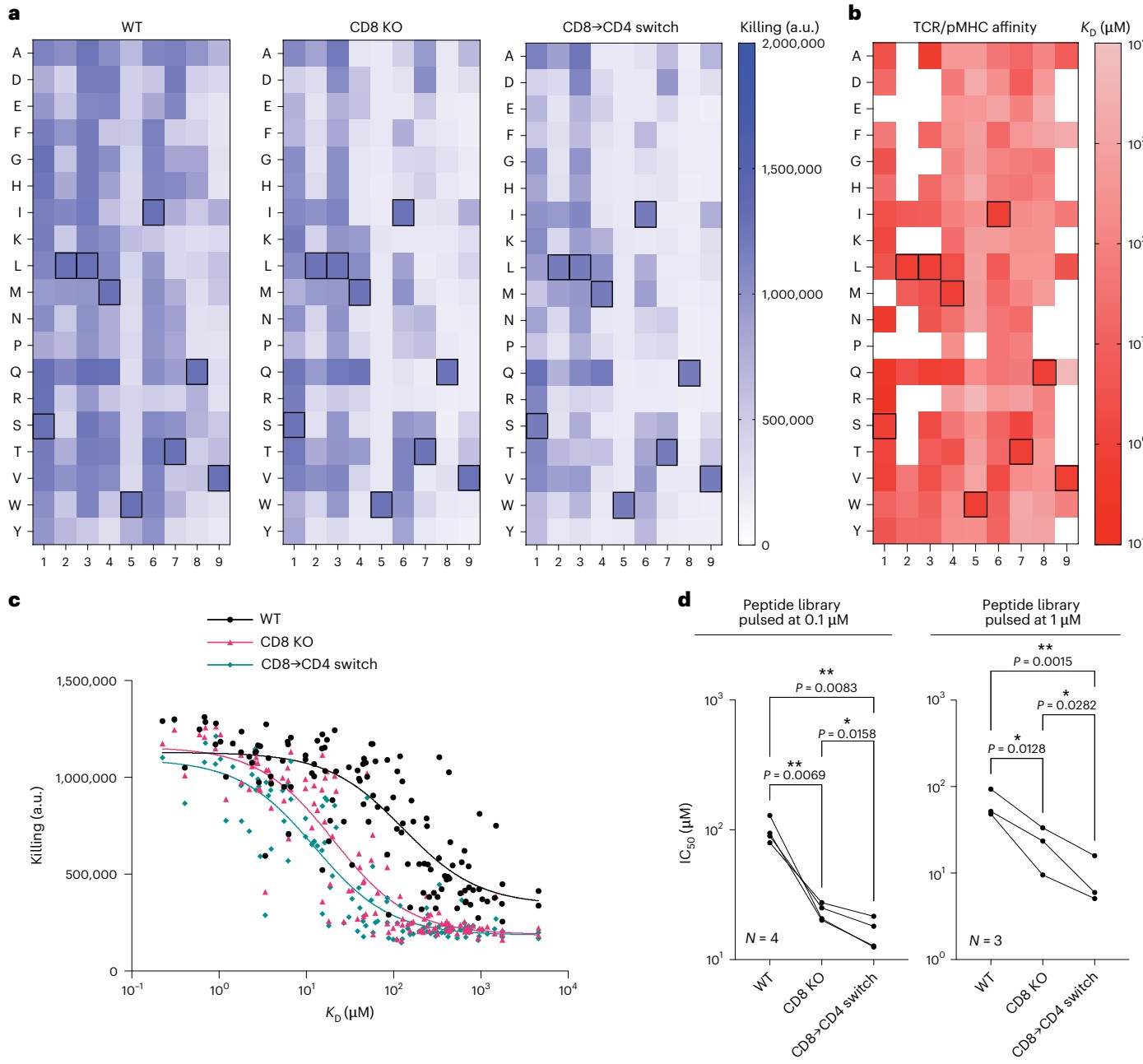

**Fig. 6 | CD8→CD4 co-receptor switch cytotoxic T cells display reduced cross-reactivity to a positional scanning peptide library. a**, U87 cells were individually loaded with 0.1 µM of each of the 163 peptides in the positional library and co-cultured with T cells. Target killing was measured after 20 h. Boxed amino acids represent the NY-ESO-1 peptide SLLMWITQV. **b**, Affinity between c259 TCR and each pMHC in the positional library determined at 37 °C by a high-throughput SPR method. Mean $K_D$ values are shown from $N = 3$ independent experiments. Boxed amino acids represent the NY-ESO-1 peptide SLLMWITQV.

White boxes represent peptides without detectable MHC binding. **c**, Target cell killing from **a** plotted over the TCR/pMHC $K_D$ from **b**. **d**, IC$_{50}$ from **c** (0.1 µM) or Supplementary Fig. 4 (1 µM) is plotted, with each data point representing an independent experiment with different blood donors. Data in **a** and **c** are representative data from $N = 4$ independent experiments with different blood donors. $P$ values were determined using one-way ANOVA and Tukey's multiple-comparisons test. *$P < 0.05$, **$P < 0.01$.

co-receptor switch T cells displayed reduced responses to target cells presenting these cross-reactive self-peptides (Fig. 7a,b and Supplementary Fig. 5). Importantly, this reduced cross-reactivity did not compromise potency to the high-affinity NY-ESO-1 9V peptide (Fig. 7c). Thus, co-receptor switching can reduce T cell cross-reactivity to increase the safety of TCR-T cell therapies.

## Discussion

It has been estimated that a single T cell can recognize over 10^6 different peptides[25,39]. This cross-reactivity is an essential feature of adaptive immunity, enabling the limited number of T cell clones within an organism to provide protection against a much larger number of pathogenic peptides. However, T cell cross-reactivity poses a substantial challenge to the success of TCR-T therapies as it can lead to lethal off-target toxicities. Identifying safe and effective TCRs remains a critical bottleneck in the development of new therapies. Despite this binding cross-reactivity, T cells use kinetic proofreading to discriminate between high and low-affinity peptides[24,34]. Since ligand discrimination emerges not only from TCR binding but also from TCR signalling[36,37], we hypothesized that modifying T cell co-signalling

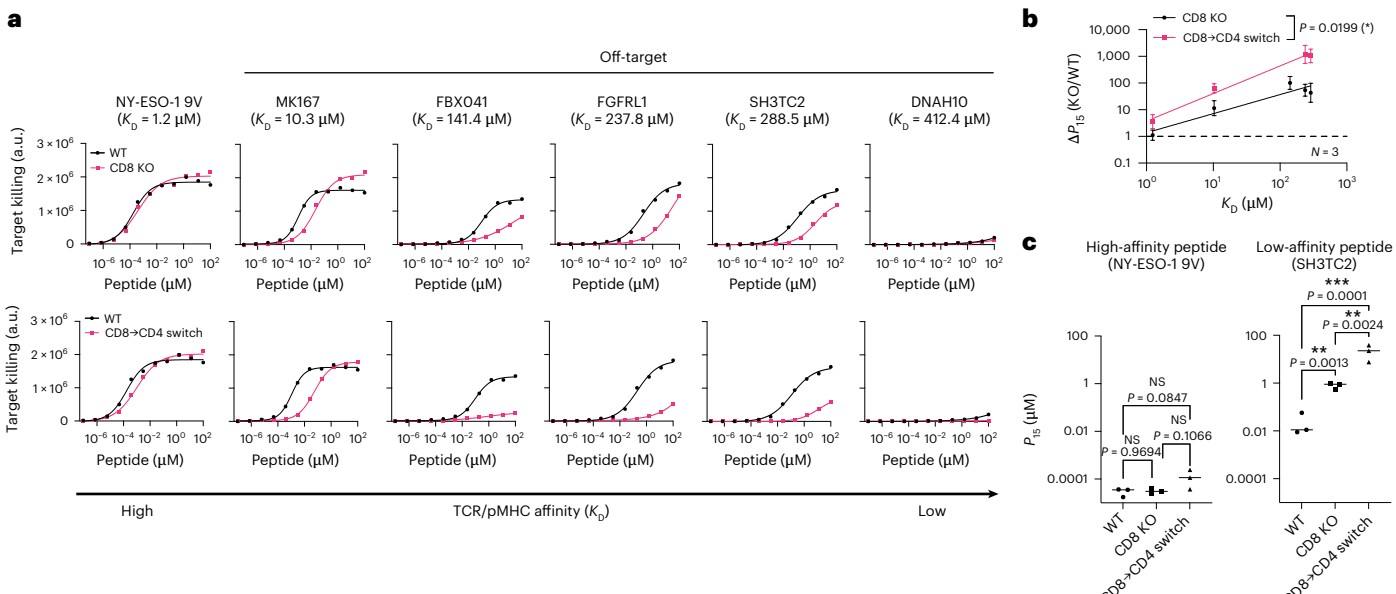

**Fig. 7 | CD8→CD4 co-receptor switch cytotoxic T cells display reduced cross-reactivity to self-peptides. a**, U87 cells were titrated with each of the predicted self-peptides to stimulate WT or CD8 KO cytotoxic c259 TCR-T cells (top), or WT or CD8→CD4 co-receptor switch cytotoxic c259 TCR-T cells (bottom). Target killing was measured after 20 h. **b**, Fold change in $P_{15}$ between modified and WT T cells from **a** is plotted over TCR–pMHC affinity ($K_D$). Data are shown as mean ± s.d. of $N = 3$ independent experiments with different blood donors. $P$ values were determined using two-tailed $F$-test. **c**, $P_{15}$ from **a** is plotted for the indicated peptides. Each data point represents an independent experiment. $P$ values were determined using one-way ANOVA and Tukey's multiple-comparisons test. Data in **a** are representative of $N = 3$ independent experiments with different blood donors. $*P < 0.05$, $**P < 0.01$, $***P < 0.001$.

receptors involved in this signalling pathway could be exploited to increase T cell ligand discrimination and reduce cross-reactivity without modifying the TCR. We have demonstrated the ability to increase and decrease ligand discrimination by genetic knockout and/or expression of the surface molecules CD5, CD4 and CD8 in helper/cytotoxic T cells. The CD8→CD4 co-receptor switch produced super selective T cells that display a striking increase in ligand discrimination and reduced cross-reactivity to pooled and positional scanning libraries, and to self-peptides without impacting on-target potency.

The CD8 co-receptor plays an essential role in thymic selection, but its role in ligand discrimination is debated. Previous work established that CD8 increases T cell activation by stabilizing the extracellular TCR–pMHC interaction[40] and by recruiting Lck to the signalling subunits of the TCR–CD3 complex[41]. It has been proposed that CD8 can selectively stabilize high-affinity TCR–pMHC interactions through a positive feedback that amplifies differences in binding affinity and hence enhances ligand discrimination[40,42]. On the other hand, it has been suggested that CD8 slows the dissociation rate of TCR–pMHC interactions[43], which preferentially increases the sensitivity to low-affinity peptides and hence reduces ligand discrimination[44–46]. Our systematic analyses support the latter hypothesis, showing that CD8 KO selectively reduces activation towards lower-affinity antigens and hence, CD8 KO increases ligand discrimination.

Our results show that the recognition of the high-affinity NY-ESO-1 9V peptide by the c259 TCR, or the recognition of the high-affinity MAGE-A3 peptide by the a3a TCR, are CD8 independent. Furthermore, we show that both TCRs are CD8 dependent for lower-affinity peptides. This is consistent with previous studies that found the threshold for co-receptor independent TCR activation to be -1 μM (ref. 47). Therefore, our findings suggest that the knockout of CD8 could be used to increase the ligand discrimination of other T cell therapies while maintaining potent target killing, if the TCR recognizes its target ligand with high affinity.

We found that CD4+ helper T cells expressing an MHC-I-restricted TCR display higher levels of ligand discrimination compared with CD8+

cytotoxic or CD8 KO cytotoxic T cells expressing the same TCR. This suggested that the CD4 co-receptor, which binds MHC-II, might further increase ligand discrimination. We confirmed this by showing that CD4 KO in helper T cells reduced their ligand discrimination, and expression of CD4 in WT or CD8 KO cytotoxic T cells enhanced their ligand discrimination. These findings are consistent with previous work[48] and with the Lck sequestration model first proposed to understand thymocyte development. This model proposes that CD4/CD8 co-receptors inhibit signalling when they are not able to recognize the ligand recognized by their TCR, impairing their TCR/ligand co-localization[49,50]. Given that co-receptors perform this function by interacting with Lck, we suggest that removal of a compatible co-receptor or the introduction of an incompatible co-receptor increases the proofreading time delay between pMHC binding and TCR signalling, leading to enhanced ligand discrimination (Fig. 3). Lastly, it is interesting to note that CD4 KO helper T cells maintained higher ligand discrimination compared with CD8 KO cytotoxic T cells: c259 TCR-T cell activation is abolished towards 4A and 5Y peptides only in the former. This suggests that other factors extrinsic to the TCR in helper T cells may endow them with higher discriminatory powers.

While increasing the discrimination of therapeutic TCRs can increase their safety, decreasing ligand discrimination has been proposed as an attractive strategy to increase activation against lower-affinity immune escape peptide variants in tumours with high genomic instability[51]. We have identified CD5 KO as a candidate modification to decrease T cell ligand discrimination and our findings are consistent with its negative regulatory function that fine-tunes TCR signalling to maintain T cell tolerance and reduce the risk of autoimmunity[26,52]. Although reducing the function of CD5 has been shown to enhance anti-tumour activity in TCR-T and CAR-T cells[53–55], this may be a double-edged sword because it would also increase cross-reactivity and hence the risk of autoimmune toxicities. Similarly, on-going clinical trials have engineered CD4+ helper T cells to express the CD8 co-receptor to increase their potency[56], but our results suggest that this may increase their cross-reactivity and the risk of autoimmune toxicities.

Overall, we have demonstrated that super selective T cells with reduced cross-reactivity and enhanced ligand discrimination can be generated without impacting on-target potency and importantly, without modifying the TCR. We have applied the method to the clinical a3a and c259 TCRs, showing that it can abolish functional cross-reactivity to self-peptides. A limitation of this method is that if a TCR does not bind its target ligand with high affinity, its potency will be reduced by co-receptor switching. Therefore, affinity maturation might be required for lower-affinity therapeutic TCRs in addition to co-receptor switching. Furthermore, the impact of co-receptor manipulations on other T cell phenotypes, such as persistence and exhaustion, will need to be assessed and optimized for the desired disease indication. Given that these super selective T cells are generated by modifying genes extrinsic to the TCR, it has the potential to dramatically increase the safety of TCR-T cell therapies using different TCRs.

## Methods

### Cell culture

U87 and HEK cell lines were cultured at 37 °C and 10% $CO_2$ in DMEM D6429 media (Sigma-Aldrich) supplemented with 10% fetal bovine serum (FBS), 50 µg ml$^{-1}$ streptomycin and 50 units ml$^{-1}$ penicillin.

T2 cells and Nalm6 cells were cultured at 37 °C and 10% $CO_2$ in RPMI 1640 (Sigma-Aldrich) supplemented with 10% FBS, 50 µg ml$^{-1}$ streptomycin and 50 units ml$^{-1}$ penicillin.

All cell lines were obtained from the ATCC except for Nalm6, which was provided by Crystal Mackall.

Primary human T cells were isolated from leucocyte cones and cultured at 37 °C and 10% $CO_2$ in RPMI 1640 (Sigma-Aldrich) supplemented with 10% FBS, 50 µg ml$^{-1}$ streptomycin, 50 units ml$^{-1}$ penicillin and 50 U ml$^{-1}$ IL-2.

### Lentivirus production

HEK 293T cells (0.8 million) were seeded in a 6-well plate (Day 1) and incubated overnight. Cells in each well were co-transfected (Day 2) using X-tremeGENE HP (Roche) with 0.8 µg of the appropriate lentiviral transfer plasmid encoding an antigen receptor (1G4 TCR or c259 TCR) and the lentiviral packaging plasmids: pRSV-Rev (0.25 µg), pMDLg/pRRE (0.53 µg) and pVSV-G (0.35 µg). The media were replaced 18 h following transfection (Day 3). At 24 h after the media exchange, the supernatant from one well was collected, filtered and used for the transduction of 1 million human T cells (Day 4).

### Production of TCR transduced primary human T cells

T cells were isolated from anonymized leucocyte cones (Day 3) purchased from the NHS Blood Donor Centre at the John Radcliffe Hospital (Oxford University Hospitals). Due to the anonymized nature of the cones, biological sex and gender were not variables in the present study and were therefore randomized, hence the authors were blinded to these variables. RosetteSep Human CD8$^+$ Enrichment Cocktail (STEMCELL Technologies) was used for cytotoxic T cells, or CD4$^+$ T Cell Enrichment Cocktail (STEMCELL Technologies) for helper T cells. The enrichment cocktail was added at 150 µl ml$^{-1}$ of sample and incubated at r.t. for 20 min. The sample was diluted with an equal volume of PBS and layered on Ficoll Paque Plus (Cytiva) density gradient medium at a 0.8:1 ratio (Ficoll:sample).

The sample was centrifuged at 1,200 g for 30 min (brake off). Cells at the interface of the Ficoll media and plasma were collected (buffy coat) and washed twice (centrifuged at 500 g for 5 min). Cells were resuspended in complete RPMI media supplemented with IL-2 (50 U ml$^{-1}$) at a density of 1 million cells per ml. Dynabeads Human T-Activator CD3/CD28 (ThermoFisher) were added (1 million beads per ml) and cells were incubated overnight.

One million cells were transduced with the filtered lentiviral supernatant (Day 4). On Day 6 and on Day 8, 1 ml of medium was removed and replaced with 1 ml of fresh medium. On Day 9, Dynabeads were removed using a magnetic stand (6 days following isolation). Cells were resuspended in fresh media every other day at a density of 1 million per ml and used for co-culture experiments. At 17 days following isolation, T cells were discarded.

### CRISPR/Cas9 knockout of T cell proteins

Cas9 ribonucleoproteins (RNPs) were prepared by mixing 8.5 µg of TruCut Cas9 protein v2 (ThermoFisher) with 150 pmol of sgRNA mix (Truguide Synthetic gRNA, ThermoFisher) for each target gene (Supplementary Table 4) and Opti-MEM (Gibco) to a final volume of 5 µl. The RNPs were incubated for 15 min at r.t.

One million freshly isolated T cells were washed with Opti-MEM (Gibco) and resuspended at a density of 20 million per ml. The T cells were mixed with the RNPs and transferred into a BTX Cuvette Plus electroporation cuvette (2 mm gap, Harvard Bioscience). The cells were electroporated using a BTX ECM 830 Square Wave Electroporation System (Harvard Bioscience) at 300 V for 2 ms. Immediately following electroporation, the cells were transferred to complete RPMI media supplemented with IL-2, and Dynabeads Human T-Activator CD3/CD28 (ThermoFisher) were added.

### Negative selection of T cell knockout cells

T cells with residual target protein expression were depleted by antibody staining and bead pulldown. T cells were resuspended in MACS buffer (PBS, 0.5% BSA, 2 mM EDTA) at a density of 100 million cells per ml. Cells were stained with 5 µl of the corresponding PE-labelled antibody per million cells for 15 min at 4 °C, washed with MACS buffer and resuspended at a density of 100 million cells per ml. A volume of 1 µl of MojoSort anti-PE nanobeads (Biolegend) was added per million cells and incubated on ice for 15 min. The cells were washed with MACS buffer and the beads were pulled down magnetically. The supernatant containing the negatively selected cells was collected.

### Cellular co-culture assays

U87 cells (50,000) in 100 µl of DMEM were seeded per well in a 96-well flat-bottom plate and incubated overnight. Alternatively, 100,000 T2 cells were placed in each well of a 96-well flat-bottom plate. Peptides were diluted in DMEM to the appropriate concentration, added to each well containing cells and incubated for 60 min at 37 °C and 10% $CO_2$. The media were discarded and 50,000 T cells were added to each well in 200 µl of RPMI medium. Cells were incubated for 20 h at 37 °C and 5% $CO_2$. Supernatants were collected for cytotoxicity and ELISA analysis. A volume of 25 µl of 100 mM EDTA PBS was added to each well containing the cells and samples were incubated for 5 min at 37 °C and 5% $CO_2$. Cells were detached by thoroughly pipetting each well and transferred to a 96-well V-bottom plate.

### Lck chemical inhibition assay

U87 cells (50,000) in 100 µl of DMEM were seeded per well in a 96-well flat-bottom plate and incubated overnight. T cells were treated with the appropriate concentration of A-770041 for 1 h. The DMEM media were discarded and 50,000 A-770041-treated T cells were added to each well in 200 µl of RPMI media. Cells were incubated for 4 h at 37 °C and 5% $CO_2$. Supernatants were collected for cytotoxicity and ELISA analysis, and T cells were analysed for activation markers as in other co-culture assays.

### Flow cytometry

The following fluorophore-conjugated mAbs were used: CD45 (Biolegend, clone HI30), CD3 (Biolegend, clone OKT3), 4-1BB (Biolegend, clone 4B4-1), CD69 (Biolegend, clone FN50), CD8α (Biolegend, clone HIT8), CD4 (Biolegend, clone RPA-T4), CD43 (Biolegend, clone CD43-10G7), CD11α (Biolegend, clone TS2/4), CD5 (Biolegend, clone UCHT2), CD2 (Biolegend, clone TS1/8) and TCR Vβ13.1 (Biolegend, clone H131).

Cells were stained for 20 min at 4 °C, washed with PBS and analysed using a BD X-20 or a Cytoflex LX flow cytometer (Beckman Couter). Data were analysed using FlowJo v.10, RRID:SCR008520 (BD Biosciences) and GraphPad Prism, RRID:SCR002798 (GraphPad Software).

## Cytotoxicity assay

Target cell lines were engineered to express the Nluc luciferase[57]. A 2 mM coelenterazine (CTZ) stock solution was prepared in methanol, aliquoted and stored at −80 °C. Supernatant from co-culture assays was mixed at a 1:1 ratio with PBS 10 µM CTZ, and luminescence was read using a SpectraMax M3 microplate reader (Molecular Devices).

## Cytokine ELISA

Invitrogen Human IFNγ ELISA kits (ThermoFisher) were used following manufacturer protocol to quantify levels of cytokine in diluted T cell supernatant. A SpectraMax M3 microplate reader (Molecular Devices) was used to measure absorbances at 450 nm and 570 nm.

## Longitudinal killing assay

mCherry positive A375 cells were seeded in a 96-well flat-bottom plate and incubated overnight in 100 µl of DMEM at 37 °C and 5% CO$_2$. To normalize differences in TCR transduction across different batches, 50,000 a3a or c259 TCR positive T cells were used as the starting concentration, and they were serially diluted to the appropriate effector:target (E:T) ratios. For each E:T ratio, 100 µl of T cells were plated in triplicate. The mCherry positive A375 cell number was quantified every 2 h using an xCELLigence RTCA eSight system (Agilent).

## Surface plasmon resonance

All SPR experiments were carried out at the Dunn School SPR facility using our published method[24]. The c259 TCR/pMHC steady-state binding affinities were measured on a Biacore T200 SPR system (GE Healthcare) with a CAP chip using HBS-EP as running buffer. The CAP chip was saturated with streptavidin and biotinylated pMHCs were immobilized to the desired level. A titration of the TCR was flowed through at 37 °C. The reference flow cell contained CD58 immobilized at levels matching those of pMHCs on the remaining flow cells. The signal from the reference flow cell was subtracted (single referencing) and the average signal from the closest buffer injection was subtracted (double referencing). Steady-state binding affinity was calculated by fitting the one site-specific binding model (Response = $B_{max}$ [TCR]/($K_D$ + [TCR])) on GraphPad Prism to double-referenced equilibrium resonance units (RU) values. The $B_{max}$ was constrained to the inferred $B_{max}$ from the empirical standard curve generated by plotting the maximal binding of a conformationally sensitive pMHC antibody to the maximal TCR binding ($B_{max}$).

## Pooled 9-mer peptide library

A library of pooled randomly synthetized 9-mer peptides was produced by Peptide Protein Research. This library was composed of all natural amino acids, except cysteine, as previously described[58]. The library has a theoretical diversity of $19^9$ peptides.

U87 cells (50,000) in 100 µl of DMEM were seeded per well in a 96-well flat-bottom plate and incubated overnight. The 9-mer pooled peptide library was diluted in DMEM to 100 µM, added to each well containing cells and incubated for 60 min at 37 °C and 10% CO$_2$. T cells (50,000) were added to each well in 200 µl of RPMI medium. Cells were incubated for 20 h at 37 °C and 5% CO$_2$. Supernatants were collected for cytotoxicity analysis. In each independent biological experiment, three technical measurements were taken and averaged.

## Positional scanning peptide library SPR

To prepare pMHC complexes presenting the local peptide library, a disulfide-stabilized variant of the human MHC-I protein HLA-A*02:01 (DS-A2) was used[59]. The DS-A2 protein was produced as described

previously[59]. Briefly, the DS-A2 and β2-microglobulin (β2m) subunits were produced in *E. coli* as inclusion bodies and solubilized in 8 M urea. The protein was then refolded in the presence of GlyLeu, a dipeptide that binds with low affinity to the peptide-binding cleft. The refolded DS-A2–β2m complexes were purified by size exclusion chromatography on a Superdex S75 10/300 column (GE Healthcare/Cytiva) in HBS-EP buffer (10 mM HEPES pH 7.4, 150 mM NaCl, 3 mM EDTA and 0.05% Tween 20). Local-library peptides were loaded by incubating the DS-A2–β2m complex with each peptide for 2 h at r.t. The pMHC complexes were stored at 4 °C until use within 24 h.

Soluble c259 TCR was produced as separate TCRα and TCRβ chains in *E. coli*. Both chains were recovered as inclusion bodies, solubilized in 100 mM Tris-HCl (pH 8.0), 8 M urea and 2 mM dithiothreitol, and then stored in aliquots at −70 °C. For refolding, 30 mg of each TCR chain was added to 1 l of refolding buffer (150 mM Tris-HCl (pH 8.0) 3 M urea, 200 mM Arg-HCl, 0.5 mM EDTA, 0.1 mM PMSF) and stirred for 1 h at 4 °C. This was followed by dialysis in 10 l 10 mM Tris-HCl (pH 8.5) buffer for 3 days in total, with the dialysis buffer changed after 1 day. The refolded c259 TCR was purified using anion exchange chromatography (HiTrap Q HP, Cytiva), followed by size exclusion chromatography (Superdex 200 Increase, Cytiva) in HBS-EP buffer. Purified c259 was used within 48 h.

High-throughput affinity measurements of c259 TCR binding to MHC loaded with the peptide library were performed using LSA or LSA$^{XT}$ (Carterra). Each pMHC was immobilized via biotin–streptavidin binding on a different spot of the SAHC30M biosensor (Carterra) for 20 min, resulting in immobilization levels between 200 and 900 RUs. Measurements were performed in HBS-EP buffer at 37 °C. A 2-fold dilution series of c259 TCR was prepared in HBS-EP buffer, with the highest concentration between 100–130 µM. Starting with the highest dilution, increasing concentrations of c259 were injected over the chip for 5 min, followed by 5–10 min of dissociation, without regeneration. Afterwards, a β2m specific antibody (clone B2M-01 (ThermoFisher) or BBM.1 (Absolute Antibody)) was injected for 10 min. The resulting data were analysed using Kinetics Software (Carterra). Any spikes were removed from the data before referencing against empty control spots or spots immobilized with CD86 at matching immobilization levels. The final injection in a series 6 buffer injections before TCR injection was subtracted from the data for double referencing. Subsequently, the steady-state binding RU was calculated by taking the average RU from over 20 s.

Steady-state analysis was performed to obtain the TCR–pMHC affinity ($K_D$) values. First, steady-state data were fitted with a one site-specific binding model (Response = $B_{max}$ [TCR]/($K_D$ + [TCR])), with $K_D$ and $B_{max}$ unconstrained. We then constructed an empirical standard curve using high-affinity pMHCs ($K_D$ < 20 µM) to relate maximal anti-β2m binding to TCR $B_{max}$. Next, steady-state data for all pMHCs were fitted with a one site-specific binding model, with $B_{max}$ constrained to the $B_{max}$ inferred from the empirical standard curve. We excluded $K_D$ values for peptides, where we observed little or no anti-β2m binding responses, indicating that the pMHC complex was unstable and lost the peptide over time (indicated as N/A in Supplementary Table 2). We further excluded $K_D$ values for pMHC that produced a TCR binding response of less than 5 RU (indicated as non-binders (NB) in Supplementary Table 2).

## Data analysis

$EC_{50}$ was calculated as the concentration of antigen required to elicit 50% of the maximum response determined for each condition individually, whereas $P_{15}$ was calculated as the concentration of antigen required to elicit 15% of the maximum activation for the experiment.

We have used $P_{15}$ for two reasons. First, $P_{15}$ always corresponds to the concentration of peptide required to activate 15% of T cells, independent of the maximum responses. In contrast, $EC_{50}$ is the concentration of peptide required to activate 50% of the maximum response

(that is, normalized to the maximum of wild type or knockout). In other words, two antigens with the same antigen potency as measured by $EC_{50}$ values may produce a different percentage of activated T cells if their maximum response ($E_{max}$) differ. In this case, the antigens would have different antigen potencies as defined by $P_{15}$. Second, the use of $P_{15}$ does not require the dose–response curve to saturate, enabling accurate estimates of $P_{15}$ from lower-affinity interactions. This measure of potency was previously used in ref. 24 to study ligand discrimination.

The study is largely focused on comparing antigen sensitivity using $EC_{50}$ or $P_{15}$ measures, which we have found to display standard deviations of 0.2 (on log-transformed values). The smallest effective size that we aimed to resolve was 3-fold changes (a difference of 0.47 on log-transformed values), and a power calculation shows that this can be resolved with a power of 80% (α at 0.05) using three samples in each group. Therefore, all experiments relied on a minimum of 3 independent donors.

### Reporting summary

Further information on research design is available in the Nature Portfolio Reporting Summary linked to this article.

## Data availability

This study has no data deposited in external repositories. Source data are provided with this paper.

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

## Acknowledgements

We thank M. Sim, Y. Samuels, S. Sagie-Groher and T. Babu for helpful discussions; J. Popplewell for assisting with the LSA SPR experiments; and Carterra Ltd for providing sensor chips. The work was funded by a Wellcome Trust Senior Fellowship in Basic Biomedical Sciences (207537/Z/17/Z to O.D.) and by UKRI-Biotechnology and Biological Sciences Research Council (BB/T008784/1 to J.C.-C.). The funders had no role in study design, data collection and analysis, decision to publish or preparation of the manuscript.

## Author contributions

J.C.-C. and O.D. conceptualized the project. J.C.-C., A.H., M.A.K., A.S. and B.W.A.P. curated data. J.C.-C. and A.H. conducted formal analysis. O.D. acquired funding. J.C.-C., A.H., M.A.K. and A.S. conducted the investigation. J.C.-C., V.A., A.H., M.A.K., A.S., B.W.A.P., G.M.G., P.A.v.d.M. and O.D. designed the methodology. O.D. administered the project. P.A.v.d.M. and O.D. supervised the project. J.C.-C. performed visualization. J.C.-C. and O.D. wrote the original manuscript draft. J.C.-C., A.H., M.A.K., A.S., P.A.v.d.M. and O.D. reviewed and edited the manuscript.

## Competing interests

J.C.-C., P.A.v.d.M. and O.D. have financial interests in a filed patent application related to this technology (UK patent application no. GB202218144D0, 2022). The remaining authors declare no competing interests.

## Ethics

For human research participants, ethics approval was provided by the Medical Sciences Inter-divisional Research Ethics Committee (IDREC) at the University of Oxford (R51997/RE001).

## Additional information

**Extended data** is available for this paper at https://doi.org/10.1038/s41551-025-01563-w.

**Correspondence and requests for materials** should be addressed to Omer Dushek.

**Reviewer recognition** *Nature Biomedical Engineering* thanks Michael Birnbaum and the other, anonymous, reviewer(s) for their contribution to the peer review of this work. Peer reviewer reports are available.

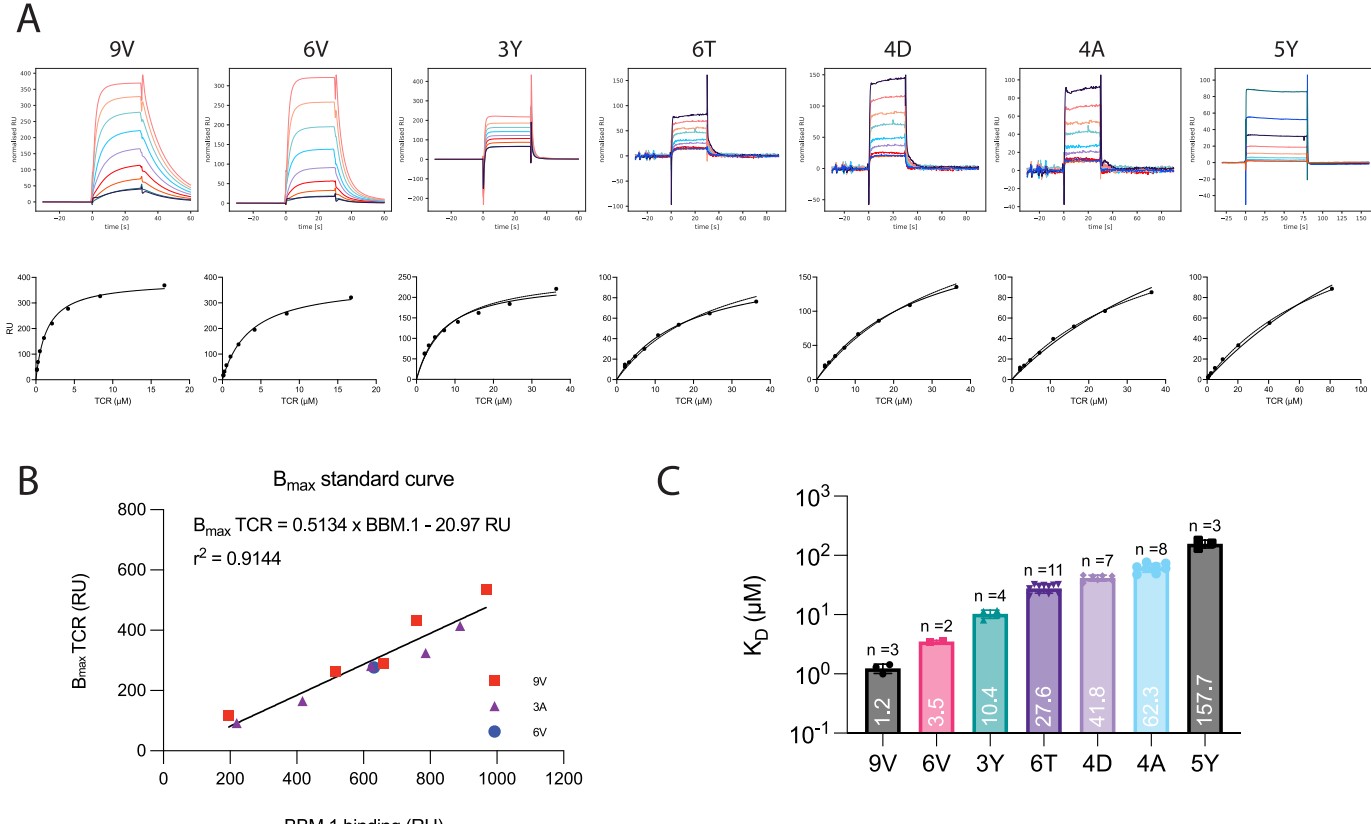

**Extended Data Fig. 1 | Establishing a panel of peptides that bind the c259 TCR with a range of affinities as measured by SPR at 37°C. (A)** (Top) Representative SPR sensograms depicting injections of increasing concentrations of the c259 TCR. (Bottom) Representative steady-state curves of c259 TCR binding to different pMHCs. 3D affinity ($K_D$) was calculated by constraining Bmax (dashed line) or fitting Bmax (solid line). **(B)** Empirical standard curve relating the binding of the BBM.1 antibody (x-axis) to the fitted TCR Bmax. Only data for the higher

affinity pMHCs is used to generate the standard curve. **(C)** Steady-state binding affinity for the selected 7-peptide panel. Barplot represents mean $K_D$ ± SDs. The affinities were calculated by constraining Bmax to the value obtained from the standard curve in **(B)** based on the amount of BBM.1 antibody that bound the chip surface (see Methods for details). All data fitting was performed using a one site-specific binding model in GraphPad Prism.

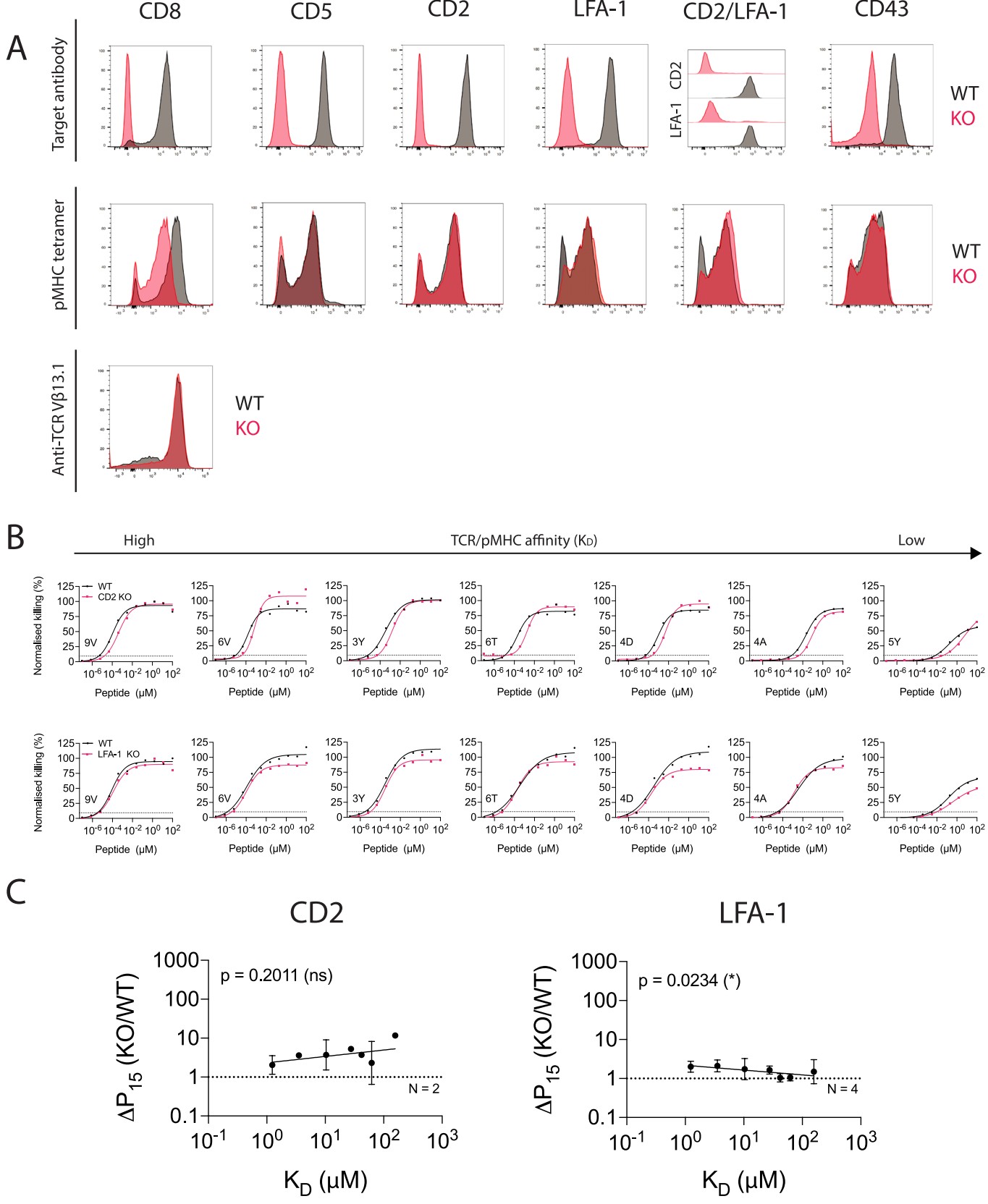

**Extended Data Fig. 2 | The impact of different T cell co-signalling receptors on ligand sensitivity and discrimination using target cell killing.** (**A**) Flow cytometry staining of WT cells (Black) or KO T cells (Red) using knock-out target antibodies, a 9V HLA-A*02:01 tetramer or an anti-TCR Vβ13.1 antibody. (**B**) U87 cells were titrated with each of the 7 NY-ESO-1 peptides to stimulate WT or KO c259 TCR-T cells. Killing of the target U87 cells was measured after

20 hours. Dashed line indicates potency (P15). (**C**) Change in potency over affinity as described in Fig. 1d. Data in (**A**) and (**B**) are representative of at least N=2 independent experiments with different blood donors. Dashed line in (**C**) indicates fold change of 1. Data in (**C**) is shown as means ± SDs. Significance of non-zero slope was assessed by a two-tailed F-test.

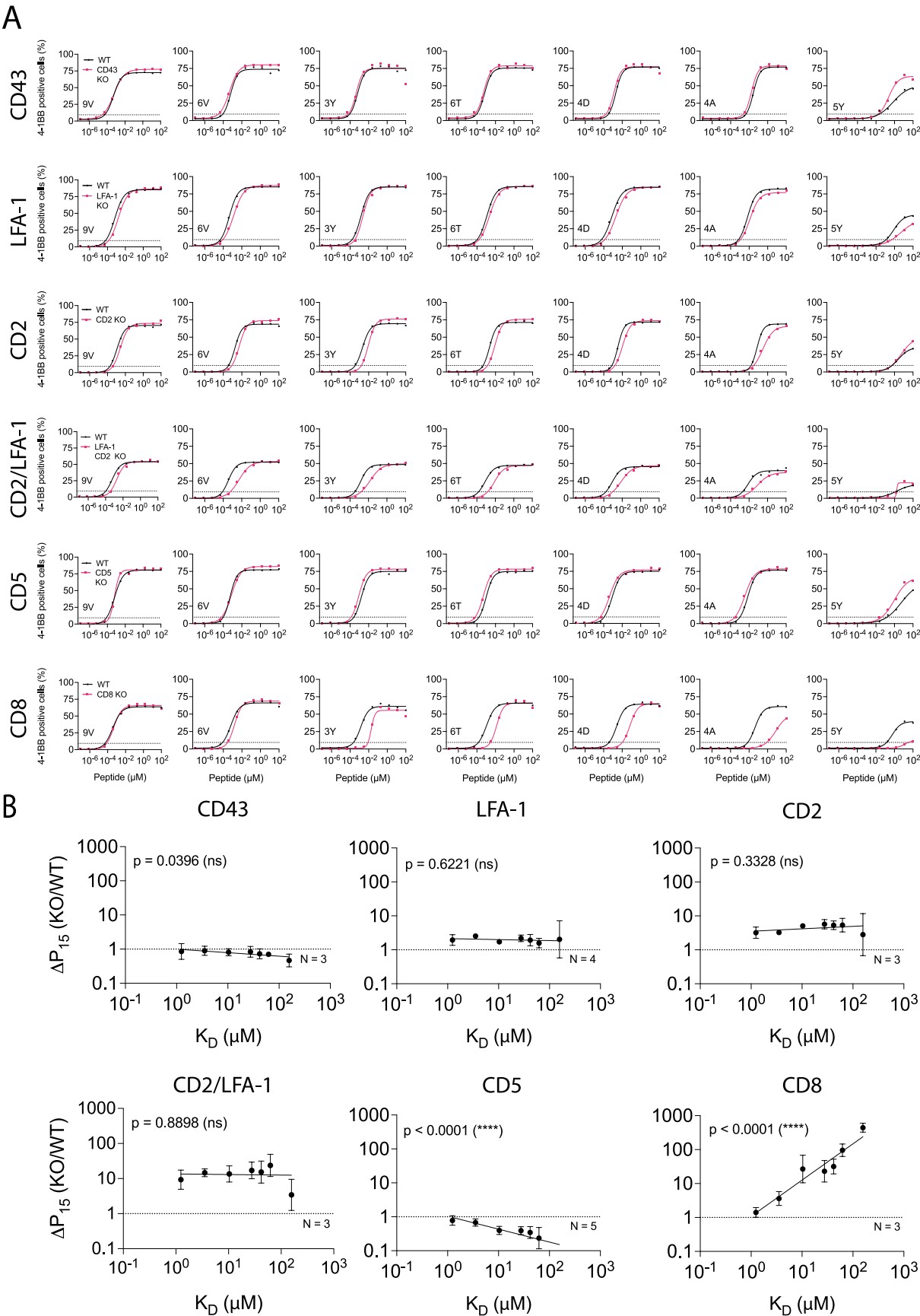

**Extended Data Fig. 3 | The impact of different T cell co-signalling receptors on ligand sensitivity and discrimination using 4-1BB activation marker.** (**A**) Representative dose-response and (**B**) Change in potency over affinity as described in Fig. 1d for target killing. Data in (**A**) are representative of at least N=3 independent experiments with different blood donors. Dashed line in (**B**) indicates fold change of 1. Data is shown as means ± SDs. Significance of non-zero slope was assessed by a two-tailed F-test.

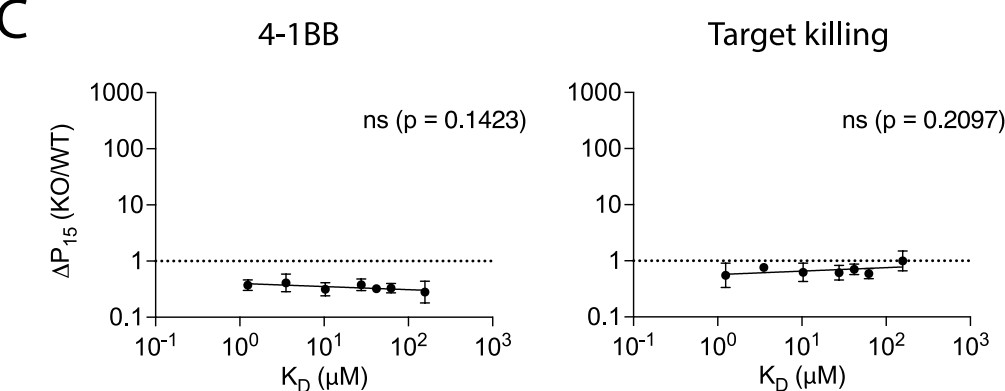

**Extended Data Fig. 4 | Knock-out of the endogenous TCR gene loci does not impact ligand discrimination.** (**A**) Flow cytometry staining of WT (Black) or TRAC/TRBC KO T cells (Red). (**B**) U87 cells were titrated with each of the 7 NY-ESO-1 peptides to stimulate WT or TRAC/TRBC KO c259 TCR-T cells. (Top) 4-1BB activation marker expression and (Bottom) target cell killing were measured. Data is representative of N=3 independent experiments with different blood donors. (**C**) Fold change in potency (P15) between TRAC/TRBC KO and WT c259-TCR T cells plotted over TCR/pMHC affinity ($K_D$). Dashed line indicates fold change of 1. Data is shown as means ± SDs of N=3 independent experiments with different blood donors. Significance of non-zero slope was assessed by a two-tailed F-test.

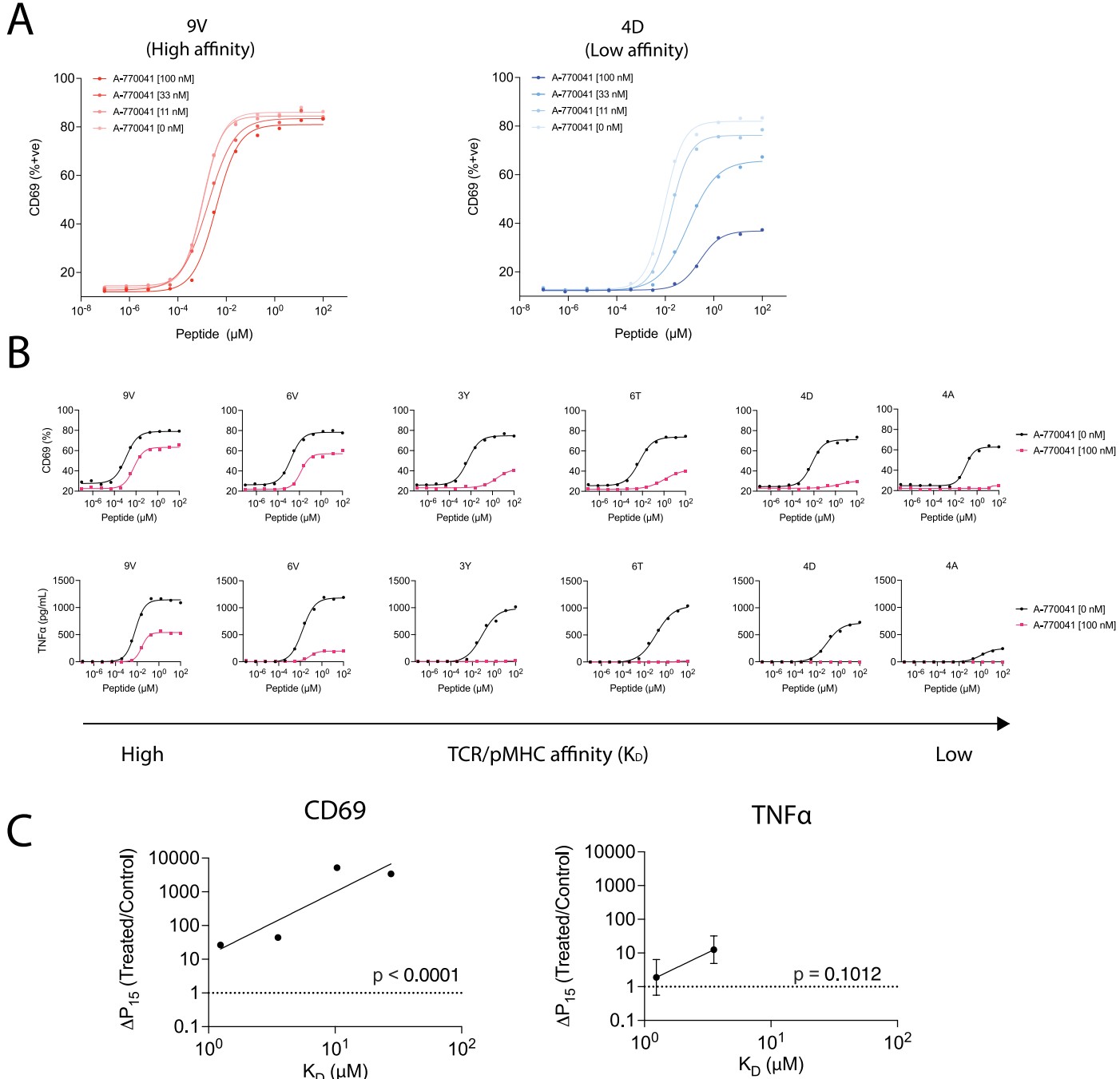

**Extended Data Fig. 5 | Chemically inhibiting Lck increases T cell ligand discrimination. (A)** U87 cells were titrated with 9V or 4D peptides to stimulate WT or A-770041 treated c259 TCR-T cells. **(B)** U87 cells were titrated with each of the 6 NY-ESO-1 peptides to stimulate WT or A-770041 treated c259 TCR-T cells. (Top) CD69 activation marker expression and (Bottom) TNFα production were measured after 4 hours. Data is representative of N=3 independent experiments with different blood donors. **(C)** Fold change in potency (P15) between A-770041 treated and WT c259 TCR-T cells plotted over TCR/pMHC affinity ($K_D$). Dashed line indicates fold change of 1. Data is shown as means ± SDs of N=3 independent experiments with different blood donors. Significance of non-zero slope was assessed by a two-tailed F-test.

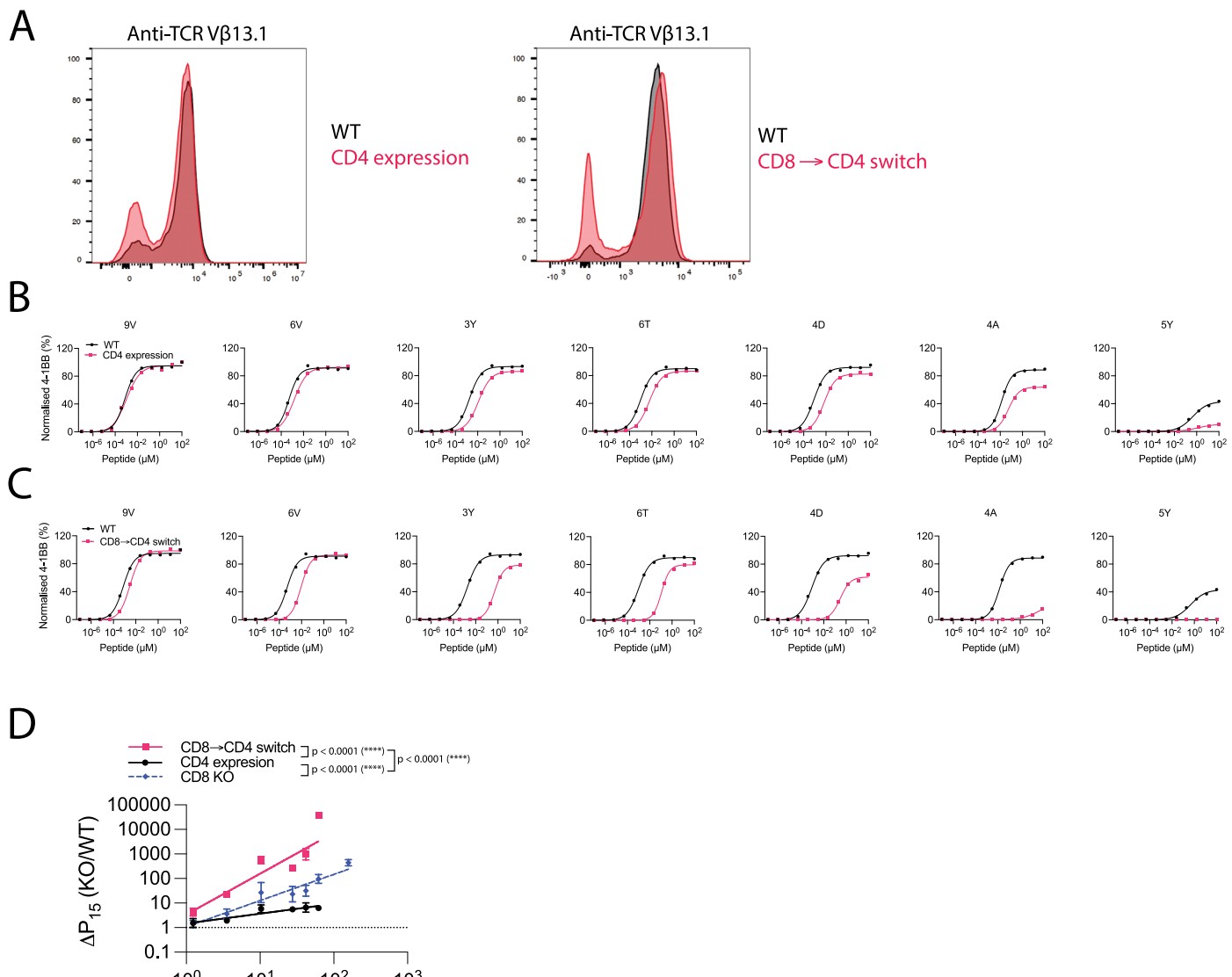

**Extended Data Fig. 6 | Expression of the incompatible CD4 co-receptor in cytotoxic T cells enhances ligand discrimination.** (**A**) Flow cytometry staining of WT cells (Black) and CD8 KO or CD8 → CD4 co-receptor switch (Red) cytotoxic c259 TCR-T cells. (**B**) U87 cells were titrated with each of the 7 NY-ESO-1 peptides to stimulate WT or CD4 expressing cytotoxic c259 TCR-T cells. 4-1BB expression was measured after 20 hours. (**C**) U87 cells were titrated with each of the 7 NY-ESO-1 peptides to stimulate WT or CD8 → CD4 co-receptor switch cytotoxic c259 TCR-T cells. 4-1BB expression was measured after 20 hours. (**D**) Fold change in potency (P15) between modified and WT T cells from (**B**,**C**) is plotted over TCR/pMHC affinity ($K_D$). Data for CD8 KO is shown from Extended Data Fig. 3. Data is shown as means ± SDs. Data in (**A**), (**B**) and (**C**) are representative of N=3 independent experiments with different blood donors. P value was determined by a two-tailed F-test. ****$p<0.0001$.

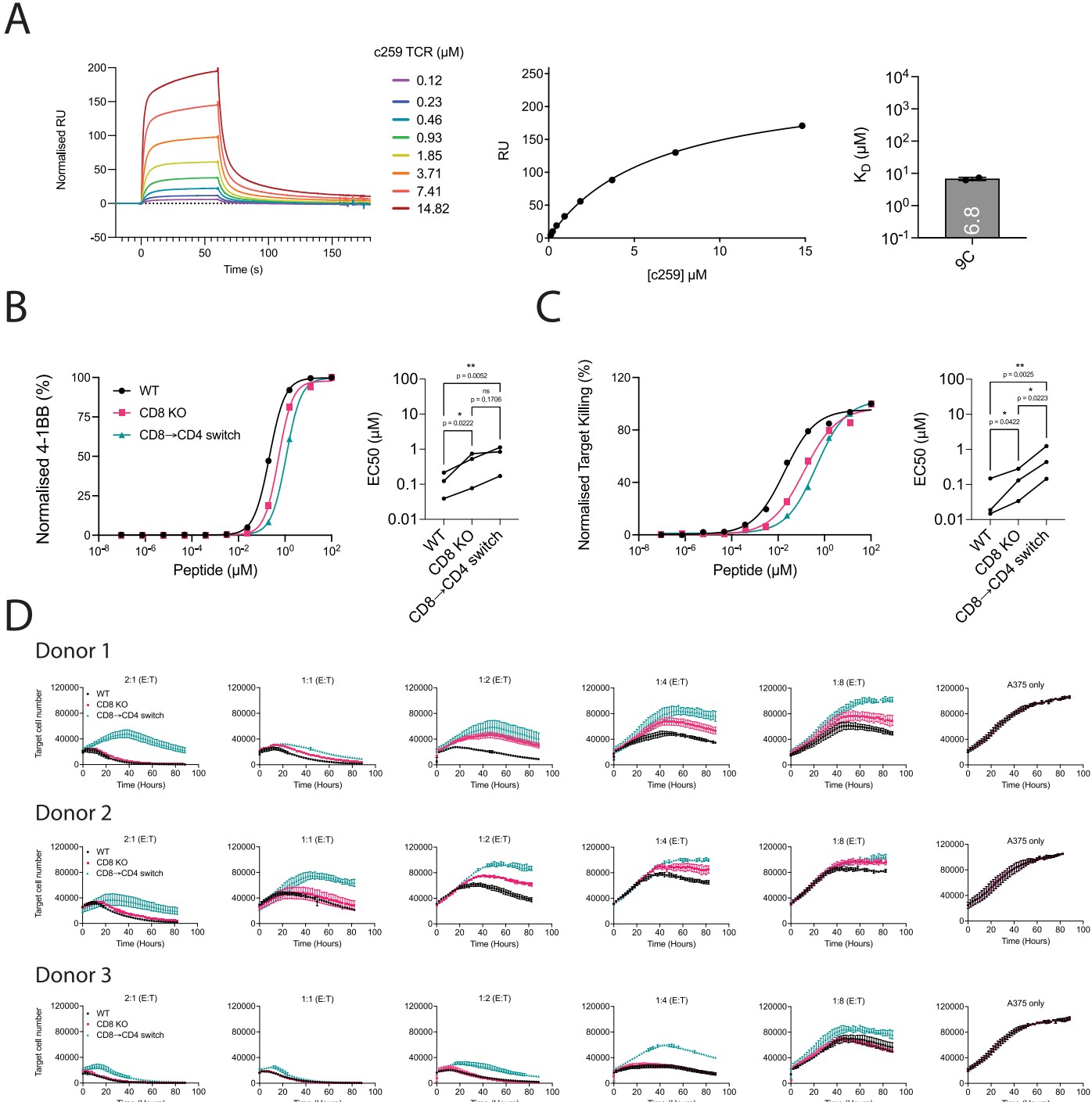

**Extended Data Fig. 7 | Activation of WT, CD8 KO and CD8 → CD4 co-receptor switch cytotoxic c259 TCR-T cells against wild-type NY-ESO-1 (9C: SLLMWITQC).** (**A**) (Left) Representative SPR sensogram depicting injections of increasing concentrations of the c259 TCR. (Middle) Representative steady-state curve of c259 TCR binding to 9C-DS-A2. 3D affinity ($K_D$) was calculated by fitting Bmax. (Right) Steady-state binding affinity. Barplot represents mean $K_D \pm$ SD from N=2 independent experiments. (**B-C**) U87 cells were titrated with the 9C NY-ESO-1 peptide to stimulate WT, CD8 KO or CD8 → CD4 co-receptor switch cytotoxic c259 TCR-T cells. (**B**) (Left) Representative 4-1BB dose-response. (Right) Summary potency (EC50). (**C**) (Left) Representative target cell killing dose-response. (Right) Summary potency (EC50). Data in left panels is representative from N=3 independent experiments. Each data point in right panels is from an independent experiment with different blood donors. P value was determined by a one-way ANOVA and Tukey's multiple-comparisons test. ns not significant, *p<0.05, **p<0.01. (**D**) A375 cells endogenously expressing the NY-ESO-1 protein were co-cultured with WT, CD8 KO or CD8 → CD4 co-receptor switch cytotoxic c259 TCR-T cells. A375 cell number was measured every two hours. Data is shown as means ± SDs of technical triplicates from N=3 independent experiments with different blood donors.

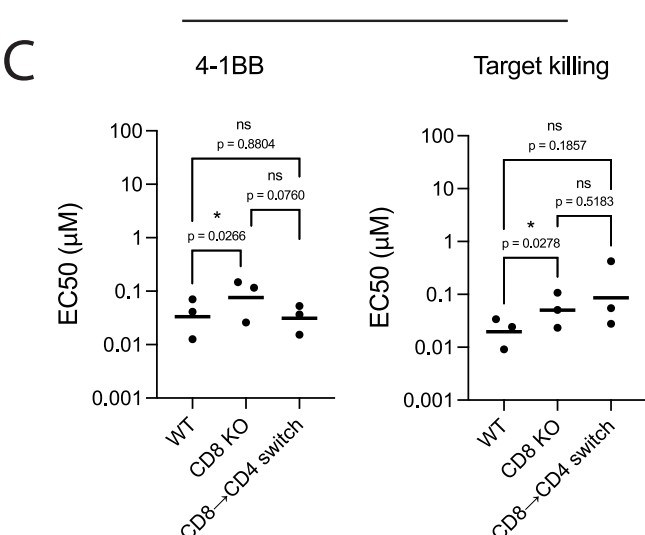

**Extended Data Fig. 8 | See next page for caption.**

**Extended Data Fig. 8 | Expression of the incompatible CD4 co-receptor enhances the ligand discrimination of 1E6-TCR T cells.** (**A**) Flow cytometry staining of 1E6 TCR (RQFGPDFPT HLA-A*02:01 tetramer), CD8 and CD4 expression. (**B**) U87 cells were titrated with four different peptides to stimulate the 1E6 TCR-T cells. (Top) 4-1BB activation marker expression or (Bottom) Killing of the target U87 cells was measured after 20 hours. Data is representative of N=3 independent experiments with different blood donors. (**C**) Potency (EC50) of WT, CD8 KO or CD8 → CD4 co-receptor switch 1E6 TCR-T cells. Each data point represents an independent experiment with different blood donors. P values were determined by a one-way ANOVA and Tukey's multiple-comparisons test. ns not significant, *p<0.05.

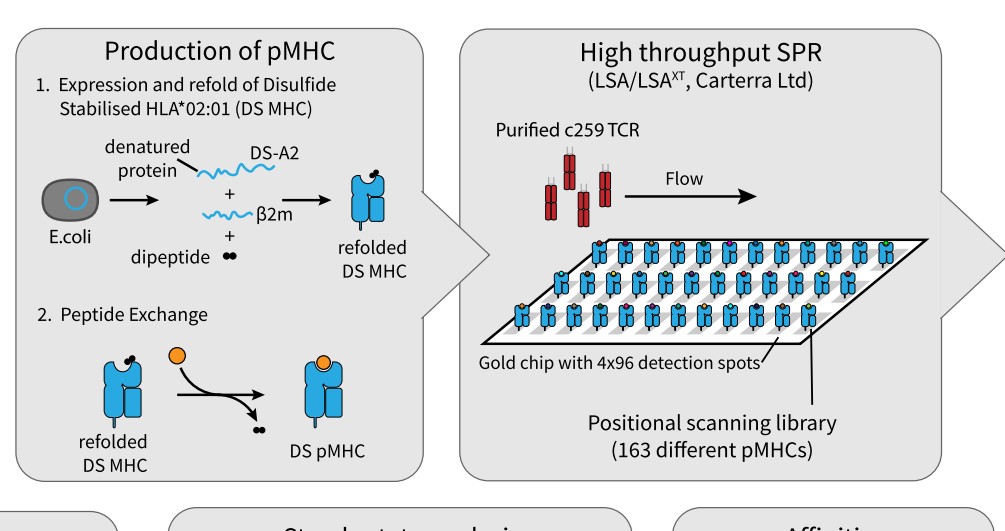

A

Workflow for parallel measurements of 163 TCR/pMHC affinities using SPR

B

C

Extended Data Fig. 9 | See next page for caption.

**Extended Data Fig. 9 | High-throughput measurements of c259 TCR affinities with the 163 pMHCs from the positional scanning library by SPR at 37°C.**
(**A**) Schematic of high-throughput SPR workflow. Step 1: Production of pMHCs presenting peptides from the positional scanning peptide library. Disulfide stabilised HLA-A*02:01 (DS-A2) and $\beta$2m are expressed in *E. coli* as denatured protein chains, then refolded with a dipeptide. The dipeptide is exchanged with a peptide from the positional scanning peptide library by incubation. Step 2: High-throughput SPR setup. Using the LSA or LSA$^{XT}$ instrument (Carterra) a pMHC carrying each peptide from the library is immobilised in a separate detection spot on the chip. Soluble TCR is injected and flows over the entire chip. Step 3: Acquisition of SPR sensograms. Each detection spot simultaneously measures TCR binding over time for each peptide from the peptide library. Step 4: Calculation of affinity values. The steady-state binding response is plotted over TCR concentration to calculate $K_D$ values using the constrained Bmax methods optimised for measuring ultra-low TCR/pMHC affinities (24). Step 5: The mean $K_D$ values as heat map. (**B**) The $K_D$ determined using the Carterra LSA/LSA$^{XT}$ instruments agrees favorably with the $K_D$ values determined using a standard BIAcore (T200). (**C**) The $K_D$ determined using the disulfide-stabilized MHC agrees favorably with the $K_D$ determined using wild-type MHC for different peptides that bind the c259 TCR with a wide range of affinities.

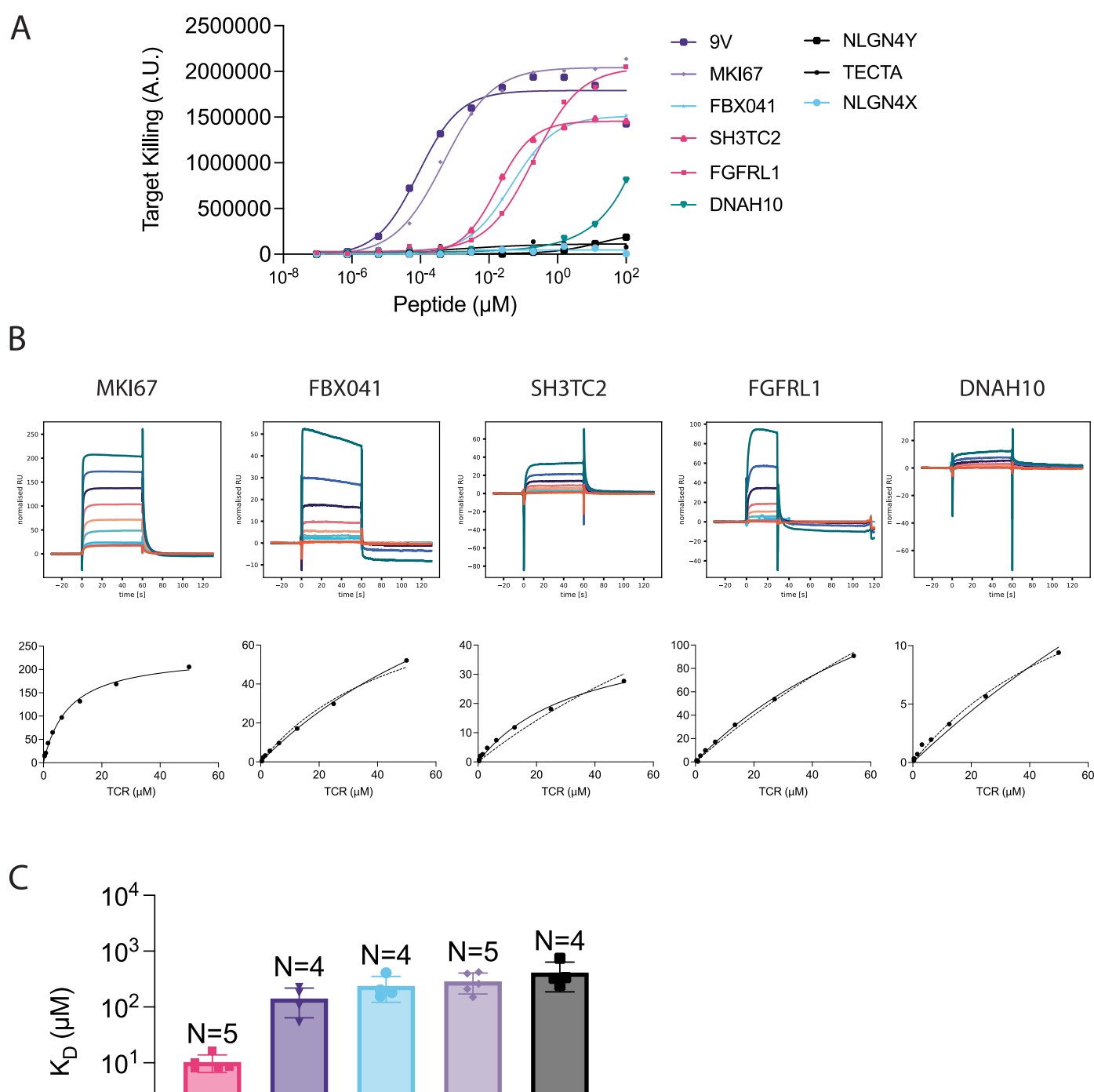

**Extended Data Fig. 10 | The c259 TCR affinity to a panel of self-peptides measured by SPR at 37°C. (A)** U87 cells were loaded with each of the listed peptides to stimulate WT c259 TCR-T cells. Target cell killing was measured after 20 hours. (**B**) The binding affinity of the c259 TCR to the peptides that induced T cell activation were measured. (Top) Representative SPR sensograms depicting injections of increasing concentrations of the c259 TCR. (Bottom) Representative equilibrium curves of c259 TCR binding to different self pMHCs. The TCR/pMHC affinity was calculated by constraining Bmax (dashed line) or fitting Bmax (solid line). (**C**) Steady-state binding affinity for the selected peptides. Barplot represents mean $K_D$ ± SDs. The affinities were calculated by constraining Bmax to the value obtained from the standard curve in (**B**) based on the amount of BBM.1 antibody that bound the chip surface (see Methods for details). All data fitting was performed using a one site-specific binding model in GraphPad Prism.

# Reporting Summary

## Statistics

For all statistical analyses, confirm that the following items are present in the figure legend, table legend, main text, or Methods section.

| n/a | Confirmed | |
|---|---|---|
| ☐ | ☒ | The exact sample size (*n*) for each experimental group/condition, given as a discrete number and unit of measurement |
| ☐ | ☒ | A statement on whether measurements were taken from distinct samples or whether the same sample was measured repeatedly |
| ☐ | ☒ | The statistical test(s) used AND whether they are one- or two-sided<br>*Only common tests should be described solely by name; describe more complex techniques in the Methods section.* |
| ☐ | ☒ | A description of all covariates tested |
| ☐ | ☒ | A description of any assumptions or corrections, such as tests of normality and adjustment for multiple comparisons |
| ☐ | ☒ | A full description of the statistical parameters including central tendency (e.g. means) or other basic estimates (e.g. regression coefficient) AND variation (e.g. standard deviation) or associated estimates of uncertainty (e.g. confidence intervals) |
| ☐ | ☒ | For null hypothesis testing, the test statistic (e.g. *F*, *t*, *r*) with confidence intervals, effect sizes, degrees of freedom and *P* value noted<br>*Give P values as exact values whenever suitable.* |
| ☒ | ☐ | For Bayesian analysis, information on the choice of priors and Markov chain Monte Carlo settings |
| ☒ | ☐ | For hierarchical and complex designs, identification of the appropriate level for tests and full reporting of outcomes |
| ☒ | ☐ | Estimates of effect sizes (e.g. Cohen's *d*, Pearson's *r*), indicating how they were calculated |

*Our web collection on statistics for biologists contains articles on many of the points above.*

## Software and code

Policy information about availability of computer code

| Data collection | No software was used for data collection. |
|---|---|
| Data analysis | FlowJo v10.10.0 (BD Biosciences) and GraphPad Prism v10.2.1 (GraphPad Software). |

For manuscripts utilizing custom algorithms or software that are central to the research but not yet described in published literature, software must be made available to editors and reviewers. We strongly encourage code deposition in a community repository (e.g. GitHub). See the Nature Portfolio guidelines for submitting code & software for further information.

## Data

Policy information about availability of data

All manuscripts must include a data availability statement. This statement should provide the following information, where applicable:
- Accession codes, unique identifiers, or web links for publicly available datasets
- A description of any restrictions on data availability
- For clinical datasets or third party data, please ensure that the statement adheres to our policy

All data is included within the manuscript and supplementary information. Source data is provided directly with the manuscript.

## Research involving human participants, their data, or biological material

Policy information about studies with human participants or human data. See also policy information about sex, gender (identity/presentation), and sexual orientation and race, ethnicity and racism.

| | |
|---|---|
| Reporting on sex and gender | This information has not been collected from the participants. |
| Reporting on race, ethnicity, or other socially relevant groupings | This information has not been collected from the participants. |
| Population characteristics | This information has not been collected from the participants. |
| Recruitment | T cells were isolated from anonymised leukocyte cones purchased from the NHS Blood Donor Centre at the John Radcliffe Hospital (Oxford University Hospitals). As a result of the anonymised nature of the cones, biological sex and gender were not variables in the present study and were therefore randomised, and as a result the authors were blinded to these variables. |
| Ethics oversight | Ethical approval was provided by the Medical Sciences Inter-divisional Research Ethics Committee (IDREC) at the University of Oxford (R51997/RE001). |

Note that full information on the approval of the study protocol must also be provided in the manuscript.

# Field-specific reporting

Please select the one below that is the best fit for your research. If you are not sure, read the appropriate sections before making your selection.

☒ Life sciences ☐ Behavioural & social sciences ☐ Ecological, evolutionary & environmental sciences

For a reference copy of the document with all sections, see nature.com/documents/nr-reporting-summary-flat.pdf

# Life sciences study design

All studies must disclose on these points even when the disclosure is negative.

| | |
|---|---|
| Sample size | Sample sizes were determined on the basis of similar published studies and of preliminary experiments. |
| Data exclusions | No data was excluded. |
| Replication | All attempts at replication were successful. |
| Randomization | No randomization was applied. |
| Blinding | No blinding was applied. |

# Reporting for specific materials, systems and methods

We require information from authors about some types of materials, experimental systems and methods used in many studies. Here, indicate whether each material, system or method listed is relevant to your study. If you are not sure if a list item applies to your research, read the appropriate section before selecting a response.

## Materials & experimental systems

| n/a | Involved in the study |
|---|---|
| ☐ | ☒ Antibodies |
| ☐ | ☒ Eukaryotic cell lines |
| ☒ | ☐ Palaeontology and archaeology |
| ☒ | ☐ Animals and other organisms |
| ☒ | ☐ Clinical data |
| ☒ | ☐ Dual use research of concern |
| ☒ | ☐ Plants |

## Methods

| n/a | Involved in the study |
|---|---|
| ☒ | ☐ ChIP-seq |
| ☐ | ☒ Flow cytometry |
| ☒ | ☐ MRI-based neuroimaging |

## Antibodies

| | |
|---|---|
| Antibodies used | CD45: Clone HI30 BV421 Biolegend RRID:AB_2561357;<br>CD3: Clone OKT3 488 Biolegend RRID:AB_571877; |

4-1BB: Clone 4B4-1 AF647 Biolegend RRID:AB_2566258;
CD69: FN50 AF647 Biolegend RRID:AB_528871;
CD8alpha: Clone HIT8 PE Biolegend RRID:AB_314112;
CD4: Clone RPA-T4 PE Biolegend RRID:AB_314075;
CD43: Clone CD43-10G7 PE Biolegend RRID:AB_2255209;
CD11alpha: Clone TS2/4 PE Biolegend RRID:AB_10660819;
CD5: Clone UCHT2 PE Biolegend RRID:AB_314094;
CD2: Clone TS1/8 PE Biolegend RRID:AB_314758;
TCR V13.1: Clone H131 APC Biolegend RRID:AB_2728348;

| Validation | All antibodies used are commercially available, and were validated by the manufacturer. |
|---|---|

# Eukaryotic cell lines

Policy information about cell lines and Sex and Gender in Research

| Cell line source(s) | U87 cell line (Source: ATCC).<br>T2 cell line (Source: ATCC).<br>Nalm6 cell line (Source: Crystal Mackall Lab).<br>A375 cell line (Source: ATCC). |
|---|---|
| Authentication | The cell lines were not authenticated. |
| Mycoplasma contamination | Cell lines tested negative for mycoplasma contamination. |
| Commonly misidentified lines<br>(See ICLAC register) | No commonly misidentified cell lines were used. |

# Plants

| Seed stocks | Report on the source of all seed stocks or other plant material used. If applicable, state the seed stock centre and catalogue number. If plant specimens were collected from the field, describe the collection location, date and sampling procedures. |
|---|---|
| Novel plant genotypes | Describe the methods by which all novel plant genotypes were produced. This includes those generated by transgenic approaches, gene editing, chemical/radiation-based mutagenesis and hybridization. For transgenic lines, describe the transformation method, the number of independent lines analyzed and the generation upon which experiments were performed. For gene-edited lines, describe the editor used, the endogenous sequence targeted for editing, the targeting guide RNA sequence (if applicable) and how the editor was applied. |
| Authentication | Describe any authentication procedures for each seed stock used or novel genotype generated. Describe any experiments used to assess the effect of a mutation and, where applicable, how potential secondary effects (e.g. second site T-DNA insertions, mosiacism, off-target gene editing) were examined. |

# Flow Cytometry

## Plots

Confirm that:

☒ The axis labels state the marker and fluorochrome used (e.g. CD4-FITC).

☒ The axis scales are clearly visible. Include numbers along axes only for bottom left plot of group (a 'group' is an analysis of identical markers).

☒ All plots are contour plots with outliers or pseudocolor plots.

☒ A numerical value for number of cells or percentage (with statistics) is provided.

## Methodology

| Sample preparation | Cells were stained in staining buffer (PBS, 1% BSA,1:200 diluted antibody) for 20 minutes at 4°C, washed with PBS and analysed. |
|---|---|
| Instrument | BD LRSFortessa X-20 cell analyzer (BD biosciences) and CytoFLEX flow cytometer (Beckman Coulter) |
| Software | FlowJo v10.10.0 (BD Biosciences) |
| Cell population abundance | Following CRISPR/Cas9 knock-out, T cells were purified using a PE-conjugated target-specific antibody and MojoSort anti-PE nanobeads. Following nanobead purification, T cell populations were characterised by staining with the target-specific antibody depending on the knock-out performed. |

Gating strategy | The starting cell population was gated on a linear SSC-A/FSC-A plot. Single cells were discriminated on a linear FSC-H/FSC-W plot. In co-culture experiments using U87 cells, T cells were gated as CD45 positive. In co-culture experiments using Nalm6 or T2 cells, T cells were gated as CD3 positive. Positive/negative populations were determined with negative controls.

☐ Tick this box to confirm that a figure exemplifying the gating strategy is provided in the Supplementary Information.

