## [Peer Review File · Nature Biomedical Engineering]

Generation of T cells with reduced off-target cross-reactivities by engineering co-signalling receptors

Corresponding Author: Prof Omer Dushek

Version 0:

Decision Letter:

Dear Omer,

Thank you again for submitting to *Nature Biomedical Engineering* your manuscript, "Generation of T cells with reduced off-target cross-reactivities by engineering co-signalling receptors", and for your patience while waiting for the full feedback. As I noted in previous correspondence, the manuscript has been seen by four experts, whose reports you will find at the end of this message (including the three reports that I had already forwarded to you).

Clearly, the reviewers appreciate the aims of the work. However, they raise concerns about the degree of support for the claims (in particular, with regards to how co-receptor modifications affect TCR responses to high-affinity targets, and the potential consequences of the presence of the endogenous TCR $\alpha\beta$), and provide useful suggestions for improvement. We hope that with substantial further work you can address the criticisms and convince the reviewers of the merits of the study. With regards to providing *in vivo* evidence, we anticipate that this can be achieved by using the same approach to engineer mouse T cells. And we assume that you will test additional TCRs, including wild-type TCRs, and TCRs recognizing mutated antigens. Importantly, please ensure that the methodology is thoroughly reported, as per the reviewers' relevant questions (such as point #2 from Reviewer #1).

When you are ready to resubmit your manuscript, please upload the revised files, a point-by-point rebuttal to the comments from all reviewers, the [reporting summary](https://www.nature.com/authors/policies/ReportingSummary.pdf), and a cover letter that explains the main improvements included in the revision and responds to any points highlighted in this decision.

Please follow the following recommendations:

- * Clearly highlight any amendments to the text and figures to help the reviewers and editors find and understand the changes (yet keep in mind that excessive marking can hinder readability).
- * If you and your co-authors disagree with a criticism, provide the arguments to the reviewer (optionally, indicate the relevant points in the cover letter).
- * If a criticism or suggestion is not addressed, please indicate so in the rebuttal to the reviewer comments and explain the reason(s).
- * Consider including responses to any criticisms raised by more than one reviewer at the beginning of the rebuttal, in a section addressed to all reviewers.
- * The rebuttal should include the reviewer comments in point-by-point format (please note that we provide all reviewers will the reports as they appear at the end of this message).
- * Provide the rebuttal to the reviewer comments and the cover letter as separate files.

We expect that you will be able to resubmit the manuscript within 20 weeks of receiving this message. If this is the case, you will be protected against potential scooping. Otherwise, we will be happy to consider a revised manuscript as long as the significance of the work is not compromised by work published elsewhere or accepted for publication at *Nature Biomedical Engineering*.

We hope that you will find the referee reports helpful when revising the work. Please do not hesitate to contact me should you have any questions.

Best wishes,

Pep

Pep Pàmies

Chief Editor, <http://www.nature.com/nbme>>Nature Biomedical Engineering

Reviewer #1 (Report for the authors (Required)):

The authors employ normal donor human TCR-T cells to investigate potential cross-reactivity to peptide ligand. The authors hypothesize that CD5, CD8, and CD4 (as well as CD43, CD2, and LFA-1) might influence ligand discrimination and use CRISPR-cas9 to delete the co-receptors of interest. Standard lentivirus transduction is used for transgene expression in T cells. The authors use a variety of in vitro methods to measure TCR-T cell activation, affinity and sensitivity to ligand presented by MHC-I. As noted, the data analysis is focused on comparing antigen sensitivity using EC50 or P15 measures. In most/all experiments, >3 independent donors are studied. Two clinically tested TCR constructs used in the study: c259 (NY-ESO-1/HLA-A*02:01) and a3a (MAGE-A3/HLA-A*01:01) represent well-established reference TCRs.

This is an interesting manuscript that reports novel findings that many immunologists would deem highly relevant for the field of cellular immunotherapies. The manuscript has many strengths including the technical competencies and rigor of the experiments described. However, a few weaknesses are noted that should be addressed. In general, the conclusions are supported by the data and have both mechanistic as well as translational significance. This study could be strengthened by addressing the following points:

1. The TCR-T cells in this study express the endogenous TCR α which may result in mixed dimers within a heterogeneous polyclonal population of mature T cells. As a result, this is a potentially confounding issue that might affect ligand discrimination. The authors should address this point experimentally to alleviate any concerns. For example, delete endogenous TRAC/TRBC in the recipient TCR-T cell population.
2. Additional information regarding the random pooled peptide library experiment in Figure 5B is required for proper interpretation. Technical details about the peptides (source, purity, number, any modifications) should be included in the materials section. Can the response of the WT, CD8 KO, and CD8-CD4 switch populations be de-convoluted in order to identify the cross-reactive ligands (in Figure 5B)?
3. It is not clear if the ligands shown in Figure 5G are "predicted" or are candidates identified from the pooled library screen. Please elaborate further.
4. p-MHC tetramer/multimer staining for each relevant TCR-T cell population would serve as an important quality control metric. For certain situations (CD8 null), staining with the TCRV β mAb would suffice.

Reviewer #2 (Report for the authors (Required)):

In the manuscript "Generation of T cells with reduced off-target cross-reactivities by engineering co-signalling receptors", Cabezas-Caballero work to create T cell receptors with improved cross-reactivity profiles through an appealingly simple means – by controlling which co-receptor is present in the T cell, and correspondingly whether the initiating kinase Lck is brought to the TCR complex (native co-receptor), excluded (switching co-receptor), or kept neutral (co-receptor knockout), the authors show that you can modulate the cross-reactivity to lower affinity peptide antigens. This work is largely well-executed and will be of interest, with some caveats/questions below:

- 1) The authors widely rely upon EC15 measurements instead of EC50 to make their point. Why is this the case? Generally, any EC50 is the standard of what is looked at, and in cases where a condition does not reach EC50, EC15 remains unreliable to calculate. With this in mind, why use EC15 instead of EC50?
- 2) Some of the peptides seem to be effected not only via potency (EC15/EC50), but by change in e_{max} (for example, the 4A and 5Y peptides in Figure 1C). Do the authors have thoughts on why this may be the case?
- 3) There appear to be differences in potency between CD4 and CD8 T cells beyond the co-receptor difference – for example, the helper T cell CD4 KO data in Figure 3H does not precisely phenocopy the cytotoxic T cell CD8 KO data in Figure 1C despite both cells at this point not having any expressed co receptors. Can the authors speculate on the origins and implications of these differences?
- 4) The authors conduct many individual measurements in the data presented in Figure 5 to show a correlation between

peptide affinity, cross-reactivity, and killing – however, I'm a bit concerned about the fact that the activity measurements represent single concentrations, which could either enhance or suppress differences depending on where the measurement is in the dose response curve. Can the authors give rationale for the use of 100nM peptide? While full dose response curves for each peptide would be ideal, it's understandable if that's an infeasible amount of data to generate. However, it would still be helpful to see (a) a dose response of the pool itself (essentially supplementing/replacing the data in Figure 5B with a titration), and/or (b) a second representative concentration datapoint.

5) It should be noted that the authors show this phenotype with two engineered TCRs – the c259 variant of 1G4 and the a3a variant of a Mage-specific TCR. On one hand, this is helpful because many of the TCRs approaching the clinic will be affinity matured. On the other hand, I think it is eminently possible – perhaps likely – that the same co-receptor manipulations would render a lower-affinity TCR inert for its target ligand. It would be helpful to determine if this is the case (using something like DMF5 TCR?), and including this caveat if so.

6) It also should be noted that this manuscript currently is completely reliant on in vitro data. While this is largely appropriate – and the mouse models are far from perfect – omission of a mouse proof of concept will be viewed upon as a weakness by at least some in the field as a necessary step for clinical translation.

7) Minor point from the discussion “We found that CD4+ helper T cells display higher levels of ligand discrimination compared to CD8+ cytotoxic or CD8 KO cytotoxic T cells recognizing MHC-I antigens.” – it should be rephrased to make it clear that the authors are claiming this for T cells expressing the same TCR, rather than a blanket statement about helper T cell ligand sensitivity vs. CTL ligand sensitivity.

Reviewer #3 (Report for the authors (Required)):

The authors very elegantly show how knockout of CD8 and overexpression of CD4 decreases the reactivity of TCRs against low-affinity MHC-peptide targets without impacting the reactivity against high-affinity targets in vitro. The manuscript is interesting, elegant, and well-written. However, major points need to be addressed before publication.

1. The authors use NYESO and MAGE-A3, both strong, CD8-independent TCRs as models. Would this approach apply to CD8-dependent TCRs? Would CD4 overexpression compensate for the CD8 requirement of CD8-dependent TCRs?

2. The authors show no impact on the reactivity of the TCRs against high-affinity MHC-peptide targets. However, the methods use high E:T ratios and only short-time reactivity. Data to support the lack of antigen-specific reactivity and potency loss is required. In vitro cytotoxicity experiments with repeated antigen stimulation (at low to medium E:T, 1:1 or 1:5), as well as in vivo antitumor studies should be conducted. These studies should use tumor models that naturally express the target antigens (avoiding forced overexpression of the targeted antigen). Additionally, conducting these studies in models of high, medium, and low antigen expression would be highly desirable. There are very limited number of tumor antigens highly and homogeneously expressed in the tumor. As a result, TCR-T cells for adoptive T cell therapy have to target tumor cells with different antigen-expression levels without impacting normal tissues.

Minor comments:

- There is a sentence that seems incomplete in lines 129 and 130.

Reviewer #4 (Report for the authors (Required)):

In this manuscript, Cabezas-Caballero et. al study the role of T cell co-receptors on ligand sensitivity for low to high affinity peptides as a strategy to engineer T cells, transduced with a therapeutic TCR that recognizes it's target with increased specificity without altering the TCR sequence itself. The authors report that out of the tested receptors CD8, CD5, CD43, CD2 and LFA-1, significant reduction to activate against lower affinity peptides, without impacting higher-affinity peptides occurred upon knockout of CD8. Further, they report that additional modulation to overexpress CD4, resulting in a CD8-to-CD4 co-receptor switch yields even more reduction in cross-reactivity. They suggest that this method of co-receptor switching can improve the safety of TCR-based immunotherapies by minimizing toxicity risks.

Major criticism:

- The concept is introduced by studying the NYESO1 reactive TCR c259 and impact of modulating co-receptors on cross-reactivity to peptide variants, while it remains unclear what the impact of co-receptor knockout or overexpression is on TCR functionality against the wild-type peptide SLLMWITQC as opposed to the peptide variants tested. Further, only data using exogenously loaded, already processed peptide is being shown. Data on how the co-receptor switch impacts T cell functionality when challenged with endogenously expressed physiological level of target antigen on HLA-A2+ and HLA-A2- cancer cell lines, such as the melanoma cell line A375 or the synovial sarcoma cell line SW982 for NYESO1 is missing.

- While the experimental strategy to study this concept in primary T cells is valid and preferred, it remains unexplored what the role of the endogenous TCR plays and whether any mispairing of endogenous with overexpressed TCR chains could impact cross-reactivity.

- CD8+ T cells recognize peptides presented in the groove of MHC Class I and the CD8 co-receptor augments avidity of this interaction by binding to non-polymorphic regions of MHC class I. One characteristic of some high potency TCRs is that they do not depend on CD8-coreceptor engagement. Such 'CD8 co-receptor independent TCRs' have distinct properties and the presented work does not investigate the role of CD8 co-receptor dependency vs. independency in this system.
- Any mechanistic insights as to what underlies this observation that CD8 KO and CD4 overexpression causes the observed effects are entirely missing. A deeper dive into the downstream TCR signaling machinery, phenotype or molecular pathways to provide a mechanistic understanding is required to advance the field.
- The authors draw broad conclusions applicable to all TCRs (line 147-149, 183-186) about ligand discrimination and preservation of reactivity against high affinity peptides, based on observations from only two affinity-matured TCRs and one parental wild-type. This is despite the fact that the data presented does not entirely support the conclusion that CD8 KO increases ligand discrimination in the parental 1G4 TCR, as data shown in Figure S5 indicates that activation is reduced both against low and moderate affinity peptides (3Y, 6T, 4D, 4A, 5Y) but also high affinity peptides (9V, 6V). Additional data showing the effects of the co-receptor switching in other TCR systems, both affinity-enhanced as well as wild-type, beyond NYESO1 and MAGE-A3 are required. In addition, the limited model systems used both target to non-mutated target antigens and at least a discussion, better an investigation of this concept in TCRs that bind mutated peptides is warranted.
- Any in vivo data, studying the potential of this engineering approach for translational impact is missing. The authors need to study the effects of co-receptor switching on T cell functionality, exhaustion, persistence in clinically relevant in vivo models of disease.

Minor comments:

- Figure 1: please clarify early in the figure legend what model system/TCR this is
- The results stated in the sentence in line 44-46 should be a Supplementary Figure
- Figure S5D: for IFN γ only 2 low concentrations were tested. More data points at higher concentrations are required.
- Figure 3: what are statistical tests were being used in each panel? Please clarify test used in the figure legend and report results (significance etc.) in the panels.

Version 1:

Decision Letter:

Dear Omer,

Thank you for your email discussing the remaining reviewer concerns with us for your manuscript "Generation of T cells with reduced off-target cross-reactivities by engineering co-signalling receptors". We agree that the paper meets field standards for assaying potency and that no further experiments are required.

Therefore, based on our editorial criteria and the referee feedback, I am pleased to write that we shall be happy to publish the manuscript in *Nature Biomedical Engineering*.

We will be performing detailed checks on your manuscript, and in due course will send you a checklist detailing our editorial and formatting requirements. You will need to follow these instructions before you upload the final manuscript files.

Best wishes,
Rita

Rita Strack, Ph.D.
Chief Editor
Nature Biomedical Engineering

Reviewer #1 (Report for the authors (Required)):

The authors have satisfactorily addressed my concerns.

Reviewer #2 (Report for the authors (Required)):

My comments have been addressed - thank you!

Reviewer #3 (Report for the authors (Required)):

The work presented is of high translational interest. It is centered on CD8+ and CD4+ T cells expressing TCRs that recognize peptides on MHC-I. Besides the model being interesting, CD4+ T cells recognizing MHC-I peptides only exist in the context of adoptive T cell therapies. As a result, this work must assess not only selectivity but also potency. A genetic manipulation that decreases cross-reactivity but also activity against the main target would not be desirable and would significantly decrease the translational relevance of this work. This is an essential question that, in my opinion, needs to be addressed in this manuscript before publication. Assays to show potency need to be added as part of the main manuscript, not in supplementary materials (this comment specifically refers to Supplementary Figure 10). Classic assays to assess potency in vitro also include sequential challenges with tumor cells. In addition, I believe that in vivo data is missing in this manuscript. The authors discuss that the FDA and EMA are moving towards reducing animal use. However, these regulatory agencies assess potential toxicities before clinical translation; they do not focus on the potential for clinical activity of the new therapeutic. While I agree that mouse models should only be used when they provide useful information, antitumor activity in vivo does provide relevant data. In addition, extensive literature supports that in vitro studies are insufficient to assess potency in cell therapies.

Reviewer #4 (Report for the authors (Required)):

This manuscript has been substantially strengthened from the initial submitted version, by clarification of previously unclear sections and the additional experimental data provided. Specifically, the studies added using Lck inhibition greatly enhance the mechanistic depth of the manuscript and additional results including TCR-KO, longitudinal killing at varying E:T ratios and inclusion of an additional non-affinity-matured TCR have strengthened the manuscript. It represents a translationally relevant study that now presents a clear and well-supported conclusion. Remaining minor suggestions include to reiterate in the discussion that for endogenously presented peptides on tumor cells, co-receptor switching may reduce potency, which should be considered for therapeutic development.

Version 2:

Decision Letter:

Dear Omer,

I am happy to inform you that your manuscript, "Generation of T cells with reduced off-target cross-reactivities by engineering co-signalling receptors", has now been accepted for publication in *Nature Biomedical Engineering*.

Over the next few weeks, the figures will be checked for production quality, the text edited to ensure that it conforms to house style, and the manuscript typeset.

You will be notified of the online publication date a few days in advance. Articles can be published any working day of the week, and are pushed live shortly after 10 am London time.

Publishing agreement. You will be asked to digitally sign a publishing agreement (grant of rights). After the signed publishing agreement has been received, the proofs of the article will be sent to you for review. If you have any queries during the production process, or you cannot meet the requested deadline for returning the proofs, please contact rjsproduction@springernature.com.

Nature Biomedical Engineering is a Transformative Journal. Authors may publish their research with us through the traditional subscription access route, or make their paper immediately open access through payment of an article-processing charge. More [information about publication options](https://www.springernature.com/gp/open-research/transformative-journals) is available.

You may need to take specific actions to [comply](https://www.springernature.com/gp/open-research/funding/policy-compliance-faqs) with funder and institutional open-access mandates. If the work described in the accepted manuscript is supported by a funder that requires immediate open access (as outlined, for example, by [Plan S](https://www.springernature.com/gp/open-research/plan-s-compliance)) and your manuscript was originally submitted on or after January 1st 2021, then you should select the gold OA route. Authors selecting subscription publication will need to accept our standard licensing terms (including our [self-archiving policies](https://www.springernature.com/gp/open-research/policies/journal-policies)), and these will supersede any other terms that the author or any third party may assert apply to any version of the manuscript.

Acceptance of your manuscript is conditional on agreement, by all authors, with both our [media embargo](http://www.nature.com/authors/policies/embargo.html) and [confidentiality and pre-publicity](http://www.nature.com/authors/policies/confidentiality.html) policies. In particular, you may arrange your own publicity of the Article (for instance, through your institutional press office), as long as you ensure that journalists strictly adhere to the media embargo.

To assist you in disseminating the work, as soon as the Article is published you will be able to take advantage of the Springer Nature [SharedIt](https://www.springernature.com/gp/researchers/sharedit) initiative to [generate a unique shareable link to the Article](http://authors.springernature.com/share) that will allow anyone (with or without a subscription) to read it. Recipients of the link who are subscribers will also be able to download and print the PDF.

Thank you for having submitted this work to *Nature Biomedical Engineering*.

Best wishes,
Rita

Rita Strack, Ph.D.
Chief Editor
Nature Biomedical Engineering
